# Determinantal point processes based on orthogonal polynomials for sampling minibatches in SGD

**Rémi Bardenet**[*]
Université de Lille, CNRS, Centrale Lille
UMR 9189 – CRIStAL, F-59000 Lille, France
`remi.bardenet@univ-lille.fr`

**Subhroshekhar Ghosh**[*][†]
National University of Singapore, Department of Mathematics
10 Lower Kent Ridge Road, 119076, Singapore
`subhrowork@gmail.com`

**Meixia Lin**[*][†]
National University of Singapore, Institute of Operations Research and Analytics
10 Lower Kent Ridge Road, 119076, Singapore
`lin_meixia@u.nus.edu`

## Abstract

Stochastic gradient descent (SGD) is a cornerstone of machine learning. When the number $N$ of data items is large, SGD relies on constructing an unbiased estimator of the gradient of the empirical risk using a small subset of the original dataset, called a minibatch. Default minibatch construction involves uniformly sampling a subset of the desired size, but alternatives have been explored for variance reduction. In particular, experimental evidence suggests drawing minibatches from determinantal point processes (DPPs), tractable distributions over minibatches that favour diversity among selected items. However, like in recent work on DPPs for coresets, providing a systematic and principled understanding of how and why DPPs help has been difficult. In this work, we contribute an orthogonal polynomial-based determinantal point process paradigm for performing minibatch sampling in SGD. Our approach leverages the specific data distribution at hand, which endows it with greater sensitivity and power over existing data-agnostic methods. We substantiate our method via a detailed theoretical analysis of its convergence properties, interweaving between the discrete data set and the underlying continuous domain. In particular, we show how specific DPPs and a string of controlled approximations can lead to gradient estimators with a variance that decays faster with the batchsize than under uniform sampling. Coupled with existing finite-time guarantees for SGD on convex objectives, this entails that, for a large enough batchsize and a fixed budget of item-level gradients to evaluate, DPP minibatches lead to a smaller bound on the mean square approximation error than uniform minibatches. Moreover, our estimators are amenable to a recent algorithm that directly samples linear statistics of DPPs (i.e., the gradient estimator) without sampling the underlying DPP (i.e., the minibatch), thereby reducing computational overhead. We provide detailed synthetic as well as real data experiments to substantiate our theoretical claims.

---

[*]Alphabetical order
[†]Corresponding author

35th Conference on Neural Information Processing Systems (NeurIPS 2021).

# 1 Introduction

Consider minimizing an empirical loss

$$\min_{\theta \in \Theta} \frac{1}{N} \cdot \sum_{i=1}^{N} \mathcal{L}(\boldsymbol{z}_i, \theta) + \lambda(\theta), \tag{1}$$

with some penalty $\lambda : \Theta \mapsto \mathbb{R}_+$. Many learning tasks, such as regression and classification, are usually framed that way [1]. When $N \gg 1$, computing the gradient of the objective in (1) becomes a bottleneck, even if individual gradients $\nabla_\theta \mathcal{L}(\boldsymbol{z}_i, \theta)$ are cheap to evaluate. For a fixed computational budget, it is thus tempting to replace vanilla gradient descent by more iterations but using an approximate gradient, obtained using only a few data points. Stochastic gradient descent (SGD; [2]) follows this template. In its simplest form, SGD builds an unbiased estimator at each iteration of gradient descent, independently from past iterations, using a minibatch of random samples from the data set. Theory [3] and practice suggest that the variance of the gradient estimators in SGD should be kept as small as possible. It is thus natural that variance reduction for SGD has been a rich area of research; see for instance the detailed references in [4, Section 2].

In a related vein, determinantal point processes (DPPs) are probability distributions over subsets of a (typically large or infinite) ground set that are known to yield samples made of collectively diverse items, while being tractable both in terms of sampling and inference. Originally introduced in electronic optics [5], they have been turned into generic statistical models for repulsion in spatial statistics [6] and machine learning [7, 8]. In ML, DPPs have also been shown to be efficient sampling tools; see Section 2. Importantly for us, there is experimental evidence that minibatches in (1) drawn from DPPs and other repulsive point processes can yield gradient estimators with low variance for advanced learning tasks [4, 9], though a conclusive theoretical result has remained elusive. In particular, it is hard to see the right candidate DPP when, unlike linear regression, the objective function does not necessarily have a geometric interpretation.

Our contributions are as follows. We combine continuous DPPs based on orthogonal polynomials [10] and kernel density estimators built on the data to obtain two gradient estimators; see Section 3. We prove that their variance is $\mathcal{O}_P(p^{-(1+1/d)})$, where $p$ is the size of the minibatch, $d$ is the dimension of data; see Section 4. This provides theoretical backing to the claim that DPPs yield variance reduction [4] over, say, uniformly sampling without replacement. In passing, the combination of analytic tools –orthogonal polynomials–, and an essentially discrete subsampling task –minibatch sampling– sheds light on new ways to build discrete DPPs for subsampling. Finally, we demonstrate our theoretical results on simulated data in Section 5.

A cornerstone of our approach is to utilise orthogonal polynomials to construct our sampling paradigm, interweaving between the discrete set of data points and the continuum in which the orthogonal polynomials reside. A few words are in order regarding the motivation for our choice of techniques. Roughly speaking, we would like to use a DPP that is tailored to the data distribution at hand. Orthogonal Polynomial Ensembles (OPEs) provide a natural way of associating a DPP to a given measure (in this case, the probability distribution of the data points), along with a substantive body of mathematical literature and tools that can be summoned as per necessity. This makes it a natural choice for our purposes.

**Notation.** Let data be denoted by $\mathfrak{D} := \{\boldsymbol{z}_1, \ldots, \boldsymbol{z}_N\}$, and assume that the $\boldsymbol{z}_i$'s are drawn i.i.d. from a distribution $\gamma$ on $\mathbb{R}^d$. Assume $\gamma$ is compactly supported, with support $\mathcal{D} \subset [-1, 1]^d$ bounded away from the border of $[-1, 1]^d$. Assume also that $\gamma$ is continuous with respect to the Lebesgue measure, and that its density is bounded away from zero on its support. While our assumptions exclude learning problems with discrete labels, such as classification, we later give experimental support that our estimators yield variance reduction in that case too. We define the *empirical measure* $\hat{\gamma}_N := N^{-1} \cdot \sum_{i=1}^{N} \delta_{\boldsymbol{z}_i}$, where $\delta_{\boldsymbol{z}_i}$ is the delta measure at the point $\boldsymbol{z}_i \in \mathbb{R}^d$. Clearly, $\hat{\gamma}_N \to \gamma$ in $\mathcal{P}(\mathbb{R}^d)$; under our operating assumption of compact support, this amounts to convergence in $\mathcal{P}(\mathcal{D})$.

For simplicity, we assume that no penalty is used in (1), but our results will extend straightforwardly. We denote the gradient of the empirical loss by $\Xi_N = \Xi_N(\theta) := N^{-1} \cdot \sum_{i=1}^{N} \nabla_\theta \mathcal{L}(\boldsymbol{z}_i, \theta)$. A minibatch is a (random) subset $A \subset [N]$ of size $|A| = p \ll N$ such that the random variable

$$\Xi_A = \Xi_A(\theta) := \sum_{i \in A} w_i \nabla_\theta \mathcal{L}(\boldsymbol{z}_i, \theta), \tag{2}$$

for suitable weights $(w_i)$, provides a good approximation for $\Xi_N$.

## 2 Background and relevant literature

**Stochastic gradient descent.** Going back to [2], SGD has been a major workhorse for machine learning; see e.g. [1, Chapter 14]. The basic version of the algorithm, applied to the empirical risk minimization (1), is to repeatedly update

$$\theta_{t+1} \leftarrow \theta_t - \eta_t \Xi_A(\theta_t), \quad t \in \mathbb{N}, \tag{3}$$

where $\eta_t$ is a (typically decreasing) stepsize, and $\Xi_A$ is a minibatch-based estimator (2) of the gradient of the empirical risk function (1), possibly depending on past iterations. Most theoretical analyses assume that for any $\theta$, $\Xi_A(\theta)$ in the $t$-th update (3) is unbiased, conditionally on the history of the Markov chain $(\theta_t)$ so far. For simplicity, we henceforth make the assumption that $\Xi_A$ does not depend on the past of the chain. In particular, using such unbiased gradients, one can derive a nonasymptotic bound [3] on the mean square distance of $\theta_t$ to the minimizer of (1), for strongly convex and smooth loss functions like linear regression and logistic regression with $\ell_2$ penalization. More precisely, for $\eta_t \propto t^{-\alpha}$ and $0 < \alpha < 1$, [3, Theorem 1] yields

$$\mathbb{E}\|\theta_t - \theta_\star\|^2 \leq f(t)\left(\mathbb{E}\|\theta_0 - \theta_\star\|^2 + \frac{\sigma^2}{L}\right) + C\frac{\sigma^2}{t^\alpha}, \tag{4}$$

where $C, L > 0$ are problem-dependent constants, $f(t) = \mathcal{O}(e^{-t^\alpha})$, and $\sigma^2 = \mathbb{E}[\|\Xi_A(\theta_\star)\|^2|\mathfrak{D}]$ is the trace of the covariance matrix of the gradient estimator, evaluated at the optimizer $\theta_\star$ of (1). The initialization bias is thus forgotten subexponentially fast, while the asymptotically leading term is proportional to $\sigma^2/t^\alpha$. Combined with practical insight that variance reduction for gradients is key, theoretical results like (4) have motivated methodological research on efficient gradient estimators [4, Section 2], i.e., constructing minibatches so as to minimize $\sigma^2$. In particular, repulsive point processes such as determinantal point processes have been empirically demonstrated to yield variance reduction and overall better performance on ML tasks [4, 9]. Our paper is a stab at a theoretical analysis to support these experimental claims.

**The Poissonian benchmark.** The default approach to sample a minibatch $A \subset [N]$ is to sample $p$ data points from $\mathfrak{D}$ uniformly, with or without replacement, and take $w_i = 1/p$ constant in (2). Both sampling with or without replacement lead to unbiased gradient estimators. A similar third approach is Poissonian random sampling. This simply consists in starting from $A = \emptyset$, and independently adding each element of $\mathfrak{D}$ to the minibatch $A$ with probability $p/N$. The Poisson estimator $\Xi_{A,\mathrm{Poi}}$ is then (2), with constant weights again equal to $1/p$. When $p \ll N$, which is the regime of SGD, the cardinality of $A$ is tightly concentrated around $p$, and $\Xi_{A,\mathrm{Poi}}$ has the same fluctuations as the two default estimators, while being easier to analyze. In particular, $\mathbb{E}[\Xi_{A,\mathrm{Poi}}|\mathfrak{D}] = \Xi_N$ and $\mathrm{Var}[\Xi_{A,\mathrm{Poi}}|\mathfrak{D}] = \mathcal{O}_P(p^{-1})$; see Appendix S1 for details.

**DPPs as (sub)sampling algorithms.** As distributions over subsets of a large ground set that favour diversity, DPPs are intuitively good candidates at subsampling tasks, and one of their primary applications in ML has been as summary extractors [7]. Since then, DPPs or mixtures thereof have been used, e.g., to generate experimental designs for linear regression, leading to strong theoretical guarantees [11, 12, 13]; see also [14] for a survey of DPPs in randomized numerical algebra, or [15] for feature selection in linear regression with DPPs.

When the objective function of the task has less structure than linear regression, it has been more difficult to prove that finite DPPs significantly improve over i.i.d. sampling. For coreset construction, for instance, [16] manages to prove that a projection DPP necessarily improves over i.i.d. sampling with the same marginals [16, Corollary 3.7], but the authors stress the disappointing fact that current concentration results for strongly Rayleigh measures (such as DPPs) do not allow yet to prove that DPP coresets need significantly fewer points than their i.i.d. counterparts [16, Section 3.2]. Even closer to our motivation, DPPs for minibatch sampling have shown promising experimental performance [4], but reading between the lines of the proof of [4, Theorem 1], a bad choice of DPP can even yield a *larger* variance than i.i.d. sampling!

For such unstructured problems (compared to, say, linear regression) as coreset extraction or loss-agnostic minibatch sampling, we propose to draw inspiration from work on continuous DPPs,

where faster-than-i.i.d. error decays have been proven in similar contexts. For instance, orthogonal polynomial theory motivated [10] to introduce a particular DPP, called a multivariate orthogonal polynomial ensemble, and prove a faster-than-i.i.d. central limit theorem for Monte Carlo integration of smooth functions. While resorting to continuous tools to study a discrete problem such as minibatch sampling may be unexpected, we shall see that the assumption that the size of the dataset is large compared to the ambient dimension crucially allows to transfer variance reduction arguments.

## 3 DPPs, and two gradient estimators

Since we shall need both discrete and continuous DPPs, we assume that $\mathcal{X}$ is either $\mathbb{R}^d$ or $\mathfrak{D}$, and follow [17] in introducing DPPs in an abstract way that encompasses both cases. After that, we propose two gradient estimators for SGD that build on a particular family of continuous DPPs introduced in [10] for Monte Carlo integration, called multivariate orthogonal polynomial ensembles.

### 3.1 DPPs: The kernel machine of point processes

A point process on $\mathcal{X}$ is a distribution on finite subsets $S$ of $\mathcal{X}$; see [18] for a general reference. Given a reference measure $\mu$ on $X$, a point process is said to be determinantal (DPP) if there exists a function $K : \mathcal{X} \times \mathcal{X} \to \mathbb{R}$, the *kernel* of the DPP, such that, for every $n \geq 1$,

$$\mathbb{E}\left[\sum_{\neq} f(x_{i_1}, \ldots, x_{i_n})\right] = \int_{(\mathcal{X})^n} f(x_1, \ldots, x_n) \cdot \mathrm{Det}\left[K(x_i, x_\ell)\right]_{i,\ell=1}^n \mathrm{d}\mu^{\otimes n}(x_1, \ldots, x_n) \quad (5)$$

for every bounded Borel function $f : \mathcal{X}^n \to \mathbb{R}$, where the sum in the LHS of (5) ranges over all pairwise distinct $n$-uplets of the random finite subset $S$. A few remarks are in order. First, satisfying (5) for every $n$ is a strong constraint on $K$, so that not every kernel yields a DPP. A set of sufficient conditions on $K$ is given by the Macchi-Soshnikov theorem [5, 19]. In words, if $K(x, y) = K(y, x)$, and if $K$ further corresponds to an integral operator

$$f \mapsto \int K(x, y) f(y) \mathrm{d}\mu(y), \quad f \in L^2(\mathcal{X}, \mu),$$

that is trace-class with spectrum in $[0, 1]$, then the corresponding DPP exists. Second, note in (5) that the kernel of a DPP encodes how the points in the random configurations interact. A strong point in favour of DPPs is that, unlike most interacting point processes, sampling and inference are tractable [17, 6]. Third, (5) yields simple formulas for the mean and variance of linear statistics of a DPP.

**Proposition 1** (See e.g. [20, 21]). *Let $S \sim DPP(K, \mu)$ and $\Phi : \mathcal{X} \to \mathbb{R}$ be a bounded Borel function,*

$$\mathbb{E}\left[\sum_{x \in S} \Phi(x)\right] = \int \Phi(x) K(x, x) \mathrm{d}\mu(x), \quad (6)$$

$$\mathrm{Var}\left[\sum_{x \in S} \Phi(x)\right] = \iint \|\Phi(x) - \Phi(y)\|_2^2 |K(x, y)|^2 \mathrm{d}\mu(x) \mathrm{d}\mu(y)$$

$$+ \int \|\Phi(x)\|_2^2 \left(K(x, x) - \int |K(x, y)|^2 \mathrm{d}\mu(y)\right) \mathrm{d}\mu(x). \quad (7)$$

Since the seminal paper [7], the case $\mathcal{X} = \mathfrak{D}$ has been the most common in machine learning. Taking $\mu$ to be $\hat{\gamma}_N$, any kernel is given by its restriction to $\mathfrak{D}$, usually given as an $N \times N$ matrix $\mathbf{K} = K|_{\mathfrak{D}}$. Equation (6) with $n = p$, $A$ a subset of size $p$ of $\mathfrak{D}$, and $\Phi$ the indicator of $A$ yields $\mathbb{P}(A \subset S) = N^{-p}\mathrm{Det}\mathbf{K}_A$. This is the usual way finite DPPs are introduced [7], except maybe for the factor $N^{-p}$, which comes from using $\hat{\gamma}_N$ as the reference measure instead of $N\hat{\gamma}_N$. In this finite setting, a careful implementation of the general DPP sampler of [17] yields $m$ DPP samples of average cardinality $\mathrm{Tr}(\mathbf{K})$ in $\mathcal{O}(N^\omega + mN\mathrm{Tr}(\mathbf{K})^2)$ operations [22].

We now go back to a general $\mathcal{X}$ and fix $p \in \mathbb{N}$. A canonical way to construct DPPs generating configurations of $p$ points almost surely, i.e. $S = \{x_1, \ldots, x_p\}$, is the following. Consider $p$ orthonormal functions $\phi_0, \ldots, \phi_{p-1}$ in $L^2(\mu)$, and take for kernel

$$K(x, y) = \sum_{k=0}^{p-1} \phi_k(x) \phi_k(y). \quad (8)$$

In this setting, the (permutation invariant) random variables $x_1, \ldots, x_p$ with joint distribution

$$\frac{1}{p!} \text{Det} \Big[ K(x_i, x_\ell) \Big]_{i,\ell=1}^p \prod_{i=1}^p \mathrm{d}\mu(x_i) \tag{9}$$

generate a DPP $\{x_1, \ldots, x_p\}$ with kernel $K(x, y)$, called a *projection* DPP. For further information on determinantal point processes, we refer the reader to [17, 7].

## 3.2  Multivariate orthogonal polynomial ensembles

This section paraphrases [10] in their definition of a particular projection DPP on $\mathcal{X} = \mathbb{R}^d$, called a multivariate orthogonal polynomial ensemble (OPE). Fix some reference measure $q(x)\mathrm{d}x$ on $[-1,1]^d$, and assume that it puts positive mass on some open subset of $[-1,1]^d$. Now choose an ordering of the monomial functions $(x_1, \ldots, x_d) \mapsto x_1^{\alpha_1} \cdots x_d^{\alpha_d}$; in this work we use the graded lexical order. Then apply the Gram-Schmidt algorithm in $L^2(q(x)\mathrm{d}x)$ to these ordered monomials. This yields a sequence of orthonormal polynomial functions $(\phi_k)_{k \in \mathbb{N}}$, the multivariate orthonormal polynomials w.r.t. $q$. Finally, plugging the first $p$ multivariate orthonormal polynomials $\phi_0, \ldots, \phi_{p-1}$ into the projection kernel (8), we obtain a projection DPP with kernel denoted as $K_q^{(p)}$, referred to as the *multivariate OPE* associated with the measure $q(x)\mathrm{d}x$.

## 3.3  Our first estimator: reweight, restrict, and saturate an OPE kernel

Let $h > 0$ and

$$\tilde{\gamma}(\boldsymbol{z}) = \frac{1}{Nh^d} \sum_{i=1}^N k\left(\frac{\boldsymbol{z} - \boldsymbol{z}_i}{h}\right) \tag{10}$$

be a single-bandwidth kernel density estimator of the pdf of the data-generating distribution $\gamma$; see [23, Section 4.2], In particular, $k$ is chosen so that $\int k(x)\mathrm{d}x = 1$. Note that the approximation kernel $k$ is unrelated to any DPP kernel in this paper. Let now $q(x) = q_1(x_1) \ldots q_d(x_d)$ be a separable pdf on $[-1,1]^d$, where each $q_i$ is Nevai-class[3]. Let $K_q^{(p)}$ be the multivariate OPE kernel defined in Section 3.2, and form a new kernel

$$K_{q,\tilde{\gamma}}^{(p)}(x,y) := \sqrt{\frac{q(x)}{\tilde{\gamma}(x)}} K_q^{(p)}(x,y) \sqrt{\frac{q(y)}{\tilde{\gamma}(y)}}.$$

The form of $K_{q,\tilde{\gamma}}^{(p)}$ is reminiscent of importance sampling [24], which is no accident. Indeed, while the positive semidefinite matrix

$$K_{q,\tilde{\gamma}}^{(p)}|_{\mathfrak{D}} := \Big( K_{q,\tilde{\gamma}}^{(p)}(\boldsymbol{z}_i, \boldsymbol{z}_j) \Big)_{1 \le i,j \le N} \tag{11}$$

is not necessarily the kernel matrix of a DPP on $(\{1, \ldots, N\}, \hat{\gamma}_N)$, see Section 3.1, we built it to be close to a projection of rank $p$. More precisely,

$$\int K_{q,\tilde{\gamma}}^{(p)}(\boldsymbol{z}_k, y) K_{q,\tilde{\gamma}}^{(p)}(y, \boldsymbol{z}_\ell)\mathrm{d}\hat{\gamma}_N(y) = \sqrt{\frac{q(\boldsymbol{z}_k)}{\tilde{\gamma}(\boldsymbol{z}_k)}} \left[ \frac{1}{N} \sum_{n=1}^N K_q^{(p)}(\boldsymbol{z}_k, \boldsymbol{z}_n) K_q^{(p)}(\boldsymbol{z}_n, \boldsymbol{z}_\ell) \frac{q(\boldsymbol{z}_n)}{\tilde{\gamma}(\boldsymbol{z}_n)} \right] \sqrt{\frac{q(\boldsymbol{z}_\ell)}{\tilde{\gamma}(\boldsymbol{z}_\ell)}}.$$

If $N$ is large compared to $p$ and $d$, so that in particular $\tilde{\gamma} \approx \gamma$, the term within brackets will be close to $\int K_q^{(p)}(\boldsymbol{z}_k, \boldsymbol{z}) K_q^{(p)}(\boldsymbol{z}, \boldsymbol{z}_\ell) q(\boldsymbol{z})\mathrm{d}\boldsymbol{z} = K_q^{(p)}(\boldsymbol{z}_k, \boldsymbol{z}_\ell)$, so that $K_{q,\tilde{\gamma}}^{(p)}|_{\mathfrak{D}}$ is almost a projection in $L^2(\hat{\gamma}_N)$.

Let us actually consider the orthogonal projection matrix $\widetilde{\mathbf{K}}$ with the same eigenvectors as $K_{q,\tilde{\gamma}}^{(p)}|_{\mathfrak{D}}$, but with the $p$ largest eigenvalues replaced by 1, and the rest of the spectrum set to 0. By the Macchi-Soshnikov theorem, $\widetilde{\mathbf{K}}$ is the kernel matrix of a DPP; see Section 3.1. We thus consider a minibatch $A \sim \text{DPP}(\widetilde{\mathbf{K}})$. Coming from a projection DPP, $|A| = p$ almost surely, and we define the gradient estimator

$$\Xi_{A,\text{DPP}} := \sum_{i \in A} \frac{\nabla_\theta \mathcal{L}(\boldsymbol{z}_i, \theta)}{\widetilde{\mathbf{K}}_{ii}}. \tag{12}$$

---

[3]See [10, Section 4] for details. It suffices that each $q_i$ is positive on $[-1,1]^d$.

In Section 4, we shall prove that $\Xi_{A,\text{DPP}}$ is unbiased, and examine under what assumptions its variance decreases faster than $1/p$.

**On the computational cost of $\Xi_{A,\text{DPP}}$.** The bottleneck is computing the $p$ largest eigenvalues of matrix (11), along with the corresponding eigenvectors. This can be done once before running SGD, as a preprocessing step. Note that storing the kernel in diagonalized form only requires $\mathcal{O}(Np)$ storage. Each iteration of SGD then only requires sampling a rank-$p$ projection DPP with diagonalized kernel, which takes $\mathcal{O}(Np^2)$ elementary operations [22]. In practice, as the complexity of the model underlying $\boldsymbol{z}, \theta \mapsto \nabla\mathcal{L}(\boldsymbol{z}, \theta)$ increases, the cost of computing $p$ individual gradients shall outweigh this $\mathcal{O}(Np^2)$ overhead. For instance, learning the parameters of a structured model like a conditional random field leads to arbitrarily costly individual gradients, as the underlying graph gets more dense [25]. Alternately, (12) can be sampled directly, without sampling the underlying DPP. Indeed the Laplace transform of (12) is a Fredholm determinant, and it is shown in [26] that Nyström-type approximations of that determinant, followed by Laplace inversion, yield an accurate inverse CDF sampler.

Finally, we stress the unusual way in which our finite DPP kernel $\widetilde{\mathbf{K}}$ is constructed, through a reweighted continuous OPE kernel, restricted to the actual dataset. This construction is interesting *per se*, as it is key to leveraging analytic techniques from the continuous case in Section 4.

### 3.4 Our second estimator: sample the OPE, but smooth the gradient

In Section 3.3, we smoothed the empirical distribution of the data and restricted a continuous kernel to the dataset $\mathfrak{D}$, to make sure that the drawn minibatch would be a subset of $\mathfrak{D}$. But one could actually define another gradient estimator, directly from an OPE sample $A = \{\boldsymbol{w}_1, \ldots, \boldsymbol{w}_p\} \sim \text{DPP}(K_q^{(p)}, q)$. Note that in that case, the "generalized minibatch" $A \subset [-1, 1]^d$ is not necessarily a subset of the dataset $\mathfrak{D}$. Defining a kernel density estimator of the gradient,

$$\widehat{\nabla_\theta \mathcal{L}}(\boldsymbol{z}, \theta) := \frac{1}{Nh^d} \sum_{i=1}^{N} \nabla_\theta \mathcal{L}(\boldsymbol{z}_i, \theta) \cdot k\left(\frac{\boldsymbol{z} - \boldsymbol{z}_i}{h}\right),$$

we consider the estimator

$$\Xi_{A,\text{s}} = \sum_{w_j \in A} \frac{\widehat{\nabla_\theta \mathcal{L}}(\boldsymbol{w}_j, \theta)}{q(\boldsymbol{w}_j) K_q^{(p)}(\boldsymbol{w}_j, \boldsymbol{w}_j)}.$$

**On the computational cost of $\Xi_{A,\text{s}}$.** Since each evaluation of this estimator is at least as costly as evaluating the actual gradient $\Xi_N(\theta)$, its use is mostly theoretical: the analysis of the fluctuations of $\Xi_{A,\text{s}}$ is easier than that of $\Xi_{A,\text{DPP}}$, while requiring the same key steps. Moreover, the computation of all pairwise distances in $\Xi_{A,\text{s}}$ could be efficiently approximated, possibly using random projection arguments [27], so that the limited scope of $\Xi_{A,\text{s}}$ might be overcome in future work. Note also that, like $\Xi_{A,\text{DPP}}$, inverse Laplace sampling [26] applies to $\Xi_{A,\text{s}}$.

## 4 Analysis of determinantal sampling of SGD gradients

We first analyze the bias, and then the fluctuations, of the gradient estimators introduced in Section 3. By Proposition 1 with $\Phi = \nabla_\theta \mathcal{L}(\cdot, \theta)$, it comes

$$\mathbb{E}[\Xi_{A,\text{DPP}}|\mathfrak{D}] = \int_D \nabla_\theta \mathcal{L}(\boldsymbol{z}, \theta) \underbrace{K_{q,\gamma}^{(p)}(\boldsymbol{z}, \boldsymbol{z})^{-1}} \cdot \underbrace{K_{q,\gamma}^{(p)}(\boldsymbol{z}, \boldsymbol{z})} \mathrm{d}\hat{\gamma}_N(z) = \frac{1}{N} \sum_{i=1}^{N} \nabla_\theta \mathcal{L}(\boldsymbol{z}_i, \theta),$$

so that we immediatly get the following result.

**Proposition 2.** $\mathbb{E}[\Xi_{A,\text{DPP}}|\mathfrak{D}] = \Xi_N.$

Thus, $\Xi_{A,\text{DPP}}$ is unbiased, like the classical Poissonian benchmark in Section 2. However, the smoothed estimator $\Xi_{A,\text{s}}$ from Section 3.4 is slightly biased. Note that while the results of [3], like (4), do not apply to biased estimators, SGD can be analyzed in the small-bias setting [28].

**Proposition 3.** *Assume that $k$ in (10) has compact support and $q$ is bounded on $D$. Then $\mathbb{E}[\Xi_{A,\text{s}}|\mathfrak{D}] = \Xi_N + \mathcal{O}_P(ph/N)$.*

*Proof.* Using the first part of Proposition 1 again, it comes

$$\mathbb{E}[\Xi_{A,\mathrm{s}}|\mathfrak{D}] = \mathbb{E}_{A\sim\mathrm{DPP}(K_q^{(p)},q)}\left[\sum_{w_j\in A}\frac{\widehat{\nabla_\theta\mathcal{L}}(w_j,\theta)}{q(w_j)K_q^{(p)}(w_j,w_j)}\right]$$

$$= \int_D \frac{\widehat{\nabla_\theta\mathcal{L}}(w,\theta)}{\cancel{q(w)}\cancel{K_q^{(p)}(w,w)}}\cdot \cancel{K_q^{(p)}(w,w)}\cancel{q(w)}\mathrm{d}w = \int_D\left(\frac{1}{Nh^d}\sum_{i=1}^N\nabla_\theta\mathcal{L}(z_i,\theta)k\left(\frac{z-z_i}{h}\right)\right)\mathrm{d}w$$

$$= \frac{1}{N}\sum_{i=1}^N\left(\int_D\frac{1}{h^d}\cdot k\left(\frac{w-z_i}{h}\right)\mathrm{d}w\right)\cdot\nabla_\theta\mathcal{L}(z_i,\theta) = \frac{1}{N}\sum_{i=1}^N\nabla_\theta\mathcal{L}(z_i,\theta) + \mathcal{O}_P(ph/N), \quad (13)$$

where, in the last step, we have used the fact that $h^{-d}\cdot k\left(h^{-1}(w-z_i)\right)$ is a probability density. The $\mathcal{O}_P(ph/N)$ error term arises because we integrate that pdf on $D$ rather than $\mathrm{Supp}(k)$. But since $k$ is a compactly supported kernel, for points $z_i$ that are within a distance $\mathcal{O}(h)$ from the boundary of $D$, we incur an error of $\mathcal{O}_P(1)$. By Proposition 1, the expected number of such points $z_i$ is $\int_{D_h}K_q^{(p)}(w,w)q(w)\mathrm{d}w$, where $D_h$ is the $h$-neighbourhood of the boundary of $D$. By a classical asymptotic result of Totik, see [10, Theorem 4.8], $w\mapsto K_q^{(p)}(w,w)$ is $\mathcal{O}(p)$ on $D_h$; whereas $q$ is a bounded density, implying that $\int_{D_h}q(w)\mathrm{d}w = \mathcal{O}(\mathrm{Vol}(D_h)) = \mathcal{O}(h)$. Putting together all of these, we obtain the $\mathcal{O}_P(ph/N)$ error term in (13). ∎

The rest of this section is devoted to the more involved task of analyzing the fluctuations of $\Xi_{A,\mathrm{DPP}}$ and $\Xi_{A,\mathrm{s}}$. For the purposes of our analysis, we discuss certain desirable regularity behaviour for our kernels and loss functions in Section 4.1. Then, we tackle the main ideas behind the study of the fluctuations of our estimators in Section 4.2. Details can be found in Appendix S3.

## 4.1  Some regularity phenomena

Assume that $(\frac{1}{p}\cdot K_q^{(p)}(z,z))^{-1}\nabla_\theta\mathcal{L}(z,\theta)$ is bounded on the domain $D$ uniformly in $N$ (note that $p$ possibly depends on $N$). Furthermore, assume that $z\mapsto (p^{-1}\cdot K_q^{(p)}(z,z)q(z))^{-1}\nabla_\theta\mathcal{L}(z,\theta)\cdot\tilde{\gamma}(z)$ is 1-Lipschitz, with Lipschitz constant $\mathcal{O}_P(1)$ and bounded in $\theta$.

Such properties are natural in the context of various known asymptotic phenomena, in particular asymptotic results on OPE kernels and the convergence of the KDE $\tilde{\gamma}$ to the distribution $\gamma$. The detailed discussion is deferred to Appendix S2; we record here in passing that the above asymptotic phenomena imply that

$$\frac{\nabla_\theta\mathcal{L}(z,\theta)}{\frac{1}{p}\cdot K_q^{(p)}(z,z)q(z)}\cdot\tilde{\gamma}(z) \longrightarrow \nabla_\theta\mathcal{L}(z,\theta)\cdot\gamma(z)\prod_{j=1}^d\sqrt{1-z[j]^2}. \quad (14)$$

Our desired regularity properties may therefore be understood in terms of closeness to this limit, which itself has similar regularity. At the price of these assumptions, we can use analytic tools to derive fluctuations for $\Xi_{A,\mathrm{DPP}}$ by working on the limit in (14). For the fluctuation analysis of the smoothed estimator $\Xi_{A,\mathrm{s}}$, we similarly assume that $(q(w)\cdot\frac{1}{p}K_q^{(p)}(w,w))^{-1}\widehat{\nabla_\theta\mathcal{L}}(w,\theta)$ is $\mathcal{O}_P(1)$ and 1-Lipschitz with an $\mathcal{O}_P(1)$ Lipschitz constant that is bounded in $\theta$. These are once again motivated by the convergence of $(q(w)\cdot\frac{1}{p}K_q^{(p)}(w,w))^{-1}\widehat{\nabla_\theta\mathcal{L}}(w,\theta)$ to $\nabla_\theta\mathcal{L}(w,\theta)\prod_{j=1}^d\sqrt{1-w[j]^2}$.

## 4.2  Reduced fluctuations for determinantal samplers

For succinctness of presentation, we focus on the theoretical analysis of the smoothed estimator $\Xi_{A,\mathrm{s}}$, leaving the analysis of $\Xi_{A,\mathrm{DPP}}$ to Appendix S3, while still capturing the main ideas of our approach.

**Proposition 4.** *Under the assumptions in Section 4.1,* $\mathrm{Var}[\Xi_{A,\mathrm{s}}|\mathfrak{D}] = \mathcal{O}_P(p^{-(1+1/d)})$.

*Proof Sketch.* Invoking (7) in the case of a projection kernel,

$$\text{Var}[\Xi_{A,s}|\mathfrak{D}] = \iint \left\| \frac{\widehat{\nabla_\theta \mathcal{L}}(\boldsymbol{z},\theta)}{q(\boldsymbol{z})K_q^{(p)}(\boldsymbol{z},\boldsymbol{z})} - \frac{\widehat{\nabla_\theta \mathcal{L}}(\boldsymbol{w},\theta)}{q(\boldsymbol{w})K_q^{(p)}(\boldsymbol{w},\boldsymbol{w})} \right\|_2^2 |K_q^{(p)}(\boldsymbol{z},\boldsymbol{w})|^2 \mathrm{d}q(\boldsymbol{z})\mathrm{d}q(\boldsymbol{w})$$

$$= \frac{1}{p^2} \iint \left\| \frac{\widehat{\nabla_\theta \mathcal{L}}(\boldsymbol{z},\theta)}{q(\boldsymbol{z}) \cdot \frac{1}{p}K_q^{(p)}(\boldsymbol{z},\boldsymbol{z})} - \frac{\widehat{\nabla_\theta \mathcal{L}}(\boldsymbol{w},\theta)}{q(\boldsymbol{w}) \cdot \frac{1}{p}K_q^{(p)}(\boldsymbol{w},\boldsymbol{w})} \right\|_2^2 |K_q^{(p)}(\boldsymbol{z},\boldsymbol{w})|^2 \mathrm{d}q(\boldsymbol{z})\mathrm{d}q(\boldsymbol{w})$$

$$\lesssim \mathcal{M}_\theta \cdot \frac{1}{p^2} \int\int \|\boldsymbol{z}-\boldsymbol{w}\|_2^2 |K_q^{(p)}(\boldsymbol{z},\boldsymbol{w})|^2 \mathrm{d}q(\boldsymbol{z})\mathrm{d}q(\boldsymbol{w}), \tag{15}$$

where we used the 1-Lipschitzianity of $\frac{\widehat{\nabla_\theta \mathcal{L}}(\boldsymbol{z},\theta)}{q(\boldsymbol{z})K_q^{(p)}(\boldsymbol{z},\boldsymbol{z})}$, with $\mathcal{M}_\theta = \mathcal{O}_P(1)$ the Lipschitz constant.

We control the integral in (15) by invoking the renowned Christoffel-Darboux formula for the OPE kernel $K_q^{(p)}$ [29]. For clarity of presentation, we outline here the main ideas for $d = 1$; the details for general dimensions are available in Appendix S3. Broadly speaking, since $K_q^{(p)}$ is an orthogonal projection of rank $p$ in $L_2(q)$, we may observe that $\iint |K_q^{(p)}(\boldsymbol{z},\boldsymbol{w})|^2 \mathrm{d}q(\boldsymbol{z})\mathrm{d}q(\boldsymbol{w}) = p$; so that without the $\|z-w\|_2^2$ term in (15), we would have a $\mathcal{O}_P(p^{-1})$ behaviour in total, which would be similar to the Poissonian estimator. However, it turns out that the leading order contribution to $\int\int \|z-w\|_2^2 |K_q^{(p)}(\boldsymbol{z},\boldsymbol{w})|^2 \mathrm{d}q(\boldsymbol{z})\mathrm{d}q(\boldsymbol{w})$ comes from near the diagonal $z = w$, and this turns out to be neutralised in an extremely precise manner by the $\|\boldsymbol{z}-\boldsymbol{w}\|_2^2$ factor that vanishes on the diagonal.

This neutralisation is systematically captured by the Christoffel-Darboux formula [29], which implies

$$K_q^{(p)}(x,y) = (x-y)^{-1} A_p \cdot (\phi_q(x)\phi_{q-1}(y) - \phi_q(y)\phi_{q-1}(x)),$$

where $A_p = \mathcal{O}(1)$ and $\phi_q, \phi_{q-1}$ are two orthonormal functions in $L_2(q)$. Substituting this into (15), a simple computation shows that $\iint \|z-w\|_2^2 |K_q^{(p)}(\boldsymbol{z},\boldsymbol{w})|^2 \mathrm{d}q(\boldsymbol{z})\mathrm{d}q(\boldsymbol{w}) = \mathcal{O}(1)$. This leads to a variance bound $\text{Var}[\Xi_{A,s}|\mathfrak{D}] = \mathcal{O}_P(p^{-2})$, which is the desired rate for $d = 1$. For general $d$, we use the fact that $q = \bigotimes_{i=1}^d q_i$ is a product measure, and apply the Christoffel-Darboux formula for each $q_i$, leading to a variance bound of $\text{Var}[\Xi_{A,s}|\mathfrak{D}] = \mathcal{O}_P(p^{-(1+1/d)})$, as desired. ∎

The theoretical analysis of $\text{Var}[\Xi_{A,\text{DPP}}|\mathfrak{D}]$ follows the broad contours of the argument for $\text{Var}[\Xi_{A,s}|\mathfrak{D}]$ as above, with additional difficulties introduced by the spectral truncation from $K_{q,\tilde{\gamma}}^{(p)}$ to $\widetilde{\mathbf{K}}$; see Section 3. This is addressed by an elaborate spectral approximation analysis in Appendix S3. In combination with the ideas expounded in the proof of Proposition 4, our analysis in Appendix S3 indicates a fluctuation bound of $\text{Var}[\Xi_A|\mathfrak{D}] = \mathcal{O}_P(p^{-(1+1/d)})$.

## 5   Experiments

In this section, we compare the empirical performance and the variance decay of our gradient estimator $\Xi_{A,\text{DPP}}$ to the default $\Xi_{A,\text{Poi}}$. We do not to include $\Xi_{A,s}$, whose interest is mostly theoretical; see Section 4. Moreover, while we focus on simulated data to illustrate our theoretical analysis, we provide experimental results on a real dataset in Appendix S4. Throughout this section, the pdf $q$ introduced in Section 3.2 is taken to be $q(\boldsymbol{w}) \propto \prod_{j=1}^d (1 + \boldsymbol{w}[j])^{\alpha_j}(1 - \boldsymbol{w}[j])^{\beta_j}$, with $\alpha_j, \beta_j \in [-1/2, 1/2]$ tuned to match the first two moments of the $j$th marginal of $\hat{\gamma}_N$. All DPPs are sampled using the Python package DPPy [30], which is under MIT licence.

**Experimental setup.**   We consider the ERM setting (1) for linear regression $\mathcal{L}_{\text{lin}}((x,y),\theta) = 0.5(\langle x,\theta \rangle - y)^2$ and logistic regression $\mathcal{L}_{\text{log}}((x,y),\theta) = \log[1 + \exp(-y\langle x,\theta \rangle)]$, both with an additional $\ell_2$ penalty $\lambda(\theta) = (\lambda_0/2)\|\theta\|_2^2$. Here the features are $x \in [-1,1]^{d-1}$, and the labels are, respectively, $y \in [-1,1]$ and $y \in \{-1,1\}$. Note that our proofs currently assume that the law $\gamma$ of $z = (x,y)$ is continuous w.r.t. Lebesgue, which in all rigour excludes logistic regression. However, we demonstrate below that if we draw a minibatch using our DPP but on features only, and then deterministically associate each label to the drawn features, we observe the same gains for logistic regression as for linear regression, where the DPP kernel takes into account both features and labels.

For each experiment, we generate $N = 1000$ data points $x_i$ with either the uniform distribution or a mixture of 2 well-separated Gaussian distributions on $[-1, 1]^{d-1}$. The variable $y_i \in \mathbb{R}$ is generated as $\langle x_i, \theta_{\text{true}} \rangle + \varepsilon_i$, where $\theta_{\text{true}} \in \mathbb{R}^{d-1}$ is given, and $\varepsilon \in \mathbb{R}^N$ is a white Gaussian noise vector. In the linear regression case, we scale $y_i$ by $\|y\|_\infty$ to make sure that $y_i \in [-1, 1]$. In the logistic regression case, we replace each $y_i$ by its sign. The regularization parameter $\lambda_0$ is manually set to be $0.1$ and the stepsize for the $t$-th iteration is set as $1/t^{0.9}$, so that (4) applies. In each experiment, performance metrics are averaged over 1000 independent runs of each SGD variant.

**Performance evaluation of sampling strategies in SGD.** Figure 1 summarizes the experimental results of $\Xi_{A,\text{Poi}}$ and $\Xi_{A,\text{DPP}}$, with $p = 5$ and $p = 10$. The top row shows how the norm of the complete gradient $\|\Xi_N(\theta_t)\|$ decreases with the number of individual gradient evaluations $t \times p$, called here *budget*. The bottom row shows the decrease of $\|\theta_t - \theta_\star\|$. Note that using $t \times p$ on all $x$-axes allows comparing different batchsizes. Error bars on the bottom row are $\pm$ one standard deviation of the mean. In all experiments, using a DPP consistently improves the performance of Poisson minibatches of the same size, and the DPP with batchsize 5 sometimes even outperforms Poisson sampling with batchsize 10, showing that smaller but more diverse batches can be a better way of spending a fixed number of gradient evaluations. This is particularly true for mixture data (middle column), where forcing diversity with our DPP brings the biggest improvement. Maybe less intuitively, the gains for the logistic regression in $d = 11$ (last column) are also significant, while the case of discrete labels is not covered yet by our theoretical analysis, and the moderately large dimension makes the improvement in the decay rate of the variance minor. This indicates that there is variance reduction beyond the change of the rate.

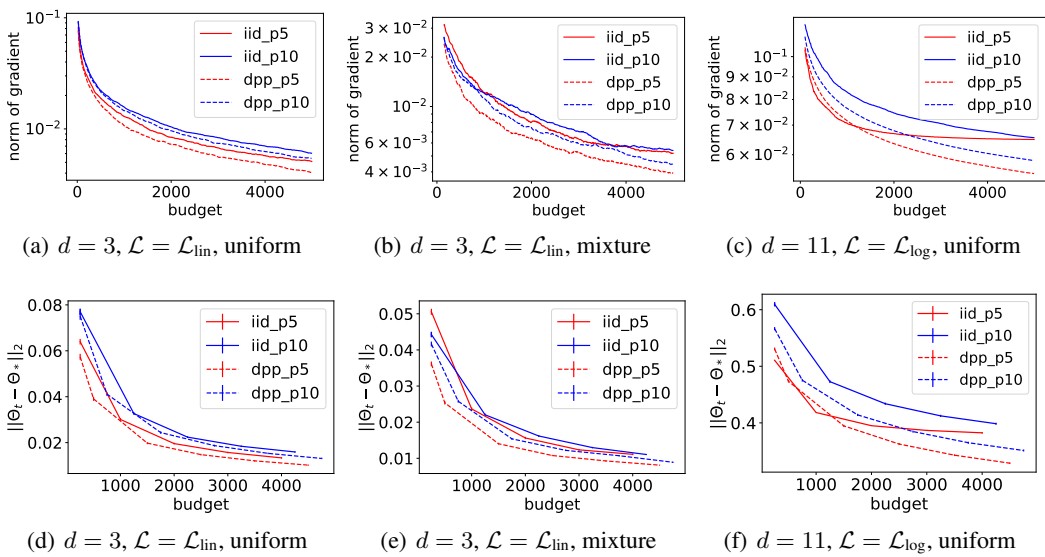

(a) $d = 3$, $\mathcal{L} = \mathcal{L}_{\text{lin}}$, uniform     (b) $d = 3$, $\mathcal{L} = \mathcal{L}_{\text{lin}}$, mixture     (c) $d = 11$, $\mathcal{L} = \mathcal{L}_{\text{log}}$, uniform

(d) $d = 3$, $\mathcal{L} = \mathcal{L}_{\text{lin}}$, uniform     (e) $d = 3$, $\mathcal{L} = \mathcal{L}_{\text{lin}}$, mixture     (f) $d = 11$, $\mathcal{L} = \mathcal{L}_{\text{log}}$, uniform

Figure 1: Summary of the performance of two sampling strategies in SGD.

**Variance decay.** For a given dimension $d$, we want to infer the rate of decay of the variance $\sigma^2 = \mathbb{E}[\|\Xi_A(\theta_\star)\|^2 | \mathfrak{D}]$, to confirm the $\mathcal{O}_P(p^{-(1+1/d)})$ rate discussed in Section 4. We take $\mathcal{L} = \mathcal{L}_{\text{lin}}$ as an example, with $N = 1000$ i.i.d. samples $\mathfrak{D}$ from the uniform distribution on $[-1, 1]^d$. For $d = 1, 2, 3$, we show in Figure 2 the sample variance of 1000 realizations of the variance of $\|\Xi_{A,\text{Poi}}(\theta_\star)\|^2$ (white dots) and $\|\Xi_{A,\text{DPP}}(\theta_\star)\|^2$ (black dots), conditionally on $\mathfrak{D}$. Blue and red dots indicate standard $95\%$ marginal confidence intervals, for indication only. The slope found by maximum likelihood in a linear regression in the log-log scale is indicated as legend. The experiment confirms that $\sigma^2$ is smaller for the DPP, decays as $p^{-1-1/d}$, and that this decay starts at a batchsize $p$ that increases with dimension.

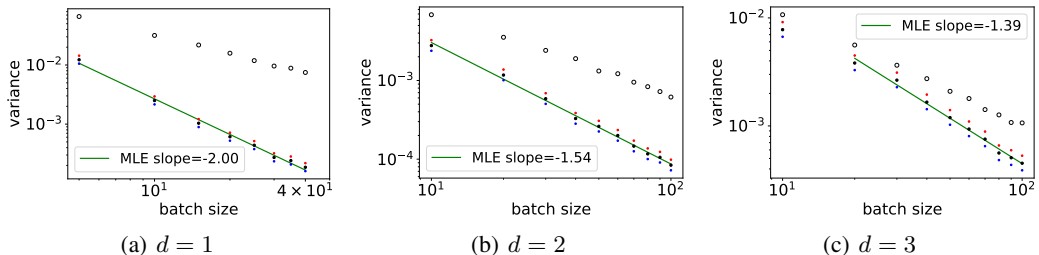

(a) $d = 1$        (b) $d = 2$        (c) $d = 3$

Figure 2: Summary of the variance decay results.

## 6 Discussion

In this work, we introduced an orthogonal polynomial-based DPP paradigm for sampling minibatches in SGD that entails variance reduction in the resulting gradient estimator. We substantiated our proposal by detailed theoretical analysis and numerical experiments. Our work raises natural questions and leaves avenues for improvement in several directions. These include the smoothed estimator $\Xi_{A,\mathrm{s}}$, which calls for further investigation in order to be deployed as a computationally attractive procedure; improvement in the dimension dependence of the fluctuation exponent when the gradients are smooth enough, like [31, 32] did for [10]; sharpening of the regularity hypotheses for our theoretical investigations to obtain a more streamlined analysis. While our estimators were motivated by a continuous underlying data distribution, our experiments suggest notably good performance in situations like logistic regression, where the data is at least partially discrete. Extensions to account for discrete settings in a principled manner, via discrete OPEs or otherwise, would be a natural topic for future research. Another natural problem is to compare our approach with other, non-i.i.d., approaches for minibatch sampling. A case in point is the method of importance sampling, where the independence across data points suggests that the variance should still be $\mathcal{O}_P(1/p)$ as in uniform sampling. More generally, incorporating ingredients from other sampling paradigms to further enhance the variance reducing capacity of our approach would be of considerable interest. Finally, while our results already partly apply to more sophisticated gradient estimators like Polyak-Rupert averaged gradients [3, Section 4.2], it would be interesting to introduce repulsiveness across consecutive SGD iterations to further minimize the variance of averaged estimators. In summary, we believe that the ideas put forward in the present work will motivate a new perspective on improved minibatch sampling for SGD, more generally on estimators based on linear statistics (e.g. in coreset sampling), and beyond.

## Acknowledgments and Disclosure of Funding

RB acknowledges support from ERC grant BLACKJACK (ERC-2019-STG-851866) and ANR AI chair BACCARAT (ANR-20-CHIA-0002). S.G. was supported in part by the MOE grants R-146-000-250-133 and R-146-000-312-114.

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
