# Supplementary material to
## *Determinantal point processes based on orthogonal polynomials for sampling minibatches in SGD*

**Rémi Bardenet**[*]
Université de Lille, CNRS, Centrale Lille
UMR 9189 – CRIStAL, F-59000 Lille, France
`remi.bardenet@univ-lille.fr`

**Subhroshekhar Ghosh**[*][†]
National University of Singapore, Department of Mathematics
10 Lower Kent Ridge Road, 119076, Singapore
`subhrowork@gmail.com`

**Meixia Lin**[*][†]
National University of Singapore, Institute of Operations Research and Analytics
10 Lower Kent Ridge Road, 119076, Singapore
`lin_meixia@u.nus.edu`

For ease of reference, sections, propositions and equations that belong to this supplementary material are prefixed with an 'S'. Additionally, labels in light blue refer to the main paper. Hyperlinks across documents should work if the two PDFs are placed in the same folder.

## S1   The Poissonian Benchmark

Our benchmark for comparing the efficacy of our estimator would be a subset $A \subset [N]$ obtained via Poissonian random sampling, which is characterised by independent random choices of the elements of $A$ from the ground set $[N]$, with each element of $[N]$ being selected independently with probability $p/N$. The estimator $\Xi_{A,\mathrm{Poi}}$ for Poissonian sampling is simply an analogue of the empirical average; in the context of (2) this is simply the choice $w_i = 1/p$ for all $i$. While the true empirical average may be realised with $w_i = 1/|A|$; we exploit here the fact that, for large $p \ll N$, the empirical cardinality $|A|$ is tightly concentrated around its expectation $p$.

Denoting by $\chi_A(\cdot)$ the indicator function for inclusion in the set $A$, the variables $\chi_A(z_i)$ are, under Poissonian sampling, i.i.d. Bernoulli random variables with parameter $p/N$. It may then be easily verified that, $\mathbb{E}[\Xi_{A,\mathrm{Poi}}|\mathfrak{D}] = \Xi_N$, whereas

$$
\begin{aligned}
\mathrm{Var}[\Xi_{A,\mathrm{Poi}}|\mathfrak{D}] &= \frac{1}{p^2} \cdot \frac{p}{N}\left(1 - \frac{p}{N}\right) \cdot \left(\sum_{i=1}^{N} \|\nabla_\theta \mathcal{L}(z_i, \theta)\|^2\right) \\
&= \frac{1}{p} \cdot \left(\frac{1}{N}\sum_{i=1}^{N} \|\nabla_\theta \mathcal{L}(z_i, \theta)\|^2\right)(1 + \mathcal{O}(1/N)).
\end{aligned} \tag{S1}
$$

---

[*]Alphabetical order
[†]Corresponding author

35th Conference on Neural Information Processing Systems (NeurIPS 2021).

Now, by a uniform central limit theorem [1], we obtain

$$\frac{1}{N}\sum_{i=1}^{N}\|\nabla_\theta \mathcal{L}(z_i,\theta)\|^2 = \int \|\nabla_\theta \mathcal{L}(z,\theta)\|^2 \mathrm{d}\gamma(z) + \mathcal{O}_P(N^{-1/2}). \tag{S2}$$

Thus, the conditional variance

$$\mathrm{Var}[\Xi_{A,\mathrm{Poi}}|\mathfrak{D}] = \left(\frac{1}{p}\cdot \int \|\nabla_\theta \mathcal{L}(z,\theta)\|^2 \mathrm{d}\gamma(z)\right) + \mathcal{O}_P(N^{-1/2}). \tag{S3}$$

where $\gamma \in \mathcal{P}(\mathbb{R}^d)$, the space of probability measures on $\mathbb{R}^d$, is the (compactly supported) distribution of the data $z$; see the paragraph on notation in Section 1. Equation (S3) provides us with the theoretical benchmark against which to compare any new approach to minibatch sampling for stochastic gradient descent.

## S2 Regularity Phenomena

For the purposes of our analysis, we envisage certain regularity behaviour for our kernels and loss functions, and discuss the natural basis for such assumptions. The discussion here complements the discussion on this topic undertaken in Section 4 of the main text.

### S2.1 Regularity phenomena and uniform CLTs

In addition to the OPE asymptotics discussed in the main text, another relevant class of asymptotic results is furnished by Glivenko-Cantelli type convergence phenomena for empirical measures and kernel density estimators. These results are naturally motivated by convergence issues arising from the law of large numbers. To elaborate, if $\{X_n\}_{n\geq 1}$ is a sequence of i.i.d. random variables following the same distribution as the random variable $X$, and $f$ is a function such that $\mathbb{E}[|f(X)|] < \infty$, then the law of large numbers tells us that the empirical average $1/N \cdot \sum_{i=1}^{N} f(X_i)$ converges almost surely to $\mathbb{E}[f(X)]$. As is also well-known, the classical central limit theorem provides the distributional order of the difference $[1/N \sum_{i=1}^{N} f(X_i) - \mathbb{E}[f(X)]]$ as $N^{-1/2}$.

But the classical central limit theorem provides only distributional information, and is not sufficient for tracing the order of such approximations as along a sequence of data $\{X_i\}_{i=1}^{N}$, as $N$ grows. Such results are rather obtained from uniform central limit theorems, which provide bounds on this error as $\mathcal{O}_P(N^{-1/2})$ that hold uniformly over large classes of functions under very mild assumptions. We refer the interested reader to the extensive literature on uniform CLTs [1].

On a related note, we would also be interested in the approximation of a measure $\nu$ with density by a kernel smoothing $\tilde{\nu}$ obtained on the basis of a dataset $\{X_i\}_{i=1}^{N}$ [2]. Analogues of the uniform CLT for such kernel density approximations are available, also according an $\mathcal{O}_P(N^{-1/2})$ approximation [3].

In our context, the uniform CLT is applicable to situations where the measure $\gamma$ is approximated by the empirical measure $\hat{\gamma}_N$, as well as $\tilde{\gamma}$ which is a kernel smoothing of $\hat{\gamma}_N$.

## S3 Fluctuation analysis for determinantal samplers

In this section, we provide the detailed proof of Proposition 4 in the main text on the fluctuations of the estimator $\Xi_{A,\mathrm{s}}$, and a theoretical analysis for the fluctuations of the estimator $\Xi_{A,\mathrm{DPP}}$. Sections S3.1.1 and S3.1.2 elaborate one of the central OPE-based ideas that enable us to obtain reduced fluctuations.

## S3.1 Detailed proof of Proposition 4

To begin with, we recall the fundamental integral expression controlling the variance of $\Xi_{A,\mathrm{s}}$, exploiting (7) in the case of a projection kernel:

$$
\mathrm{Var}[\Xi_{A,\mathrm{s}}|\mathfrak{D}] = \iint \left\| \frac{\widehat{\nabla_\theta \mathcal{L}}(\boldsymbol{z},\theta)}{q(\boldsymbol{z})K_q^{(p)}(\boldsymbol{z},\boldsymbol{z})} - \frac{\widehat{\nabla_\theta \mathcal{L}}(\boldsymbol{w},\theta)}{q(\boldsymbol{w})K_q^{(p)}(\boldsymbol{w},\boldsymbol{w})} \right\|_2^2 |K_q^{(p)}(\boldsymbol{z},\boldsymbol{w})|^2 \mathrm{d}q(\boldsymbol{z})\mathrm{d}q(\boldsymbol{w})
$$

$$
= \frac{1}{p^2} \iint \left\| \frac{\widehat{\nabla_\theta \mathcal{L}}(\boldsymbol{z},\theta)}{q(\boldsymbol{z}) \cdot \frac{1}{p}K_q^{(p)}(\boldsymbol{z},\boldsymbol{z})} - \frac{\widehat{\nabla_\theta \mathcal{L}}(\boldsymbol{w},\theta)}{q(\boldsymbol{w}) \cdot \frac{1}{p}K_q^{(p)}(\boldsymbol{w},\boldsymbol{w})} \right\|_2^2 |K_q^{(p)}(\boldsymbol{z},\boldsymbol{w})|^2 \mathrm{d}q(\boldsymbol{z})\mathrm{d}q(\boldsymbol{w})
$$

$$
\lesssim \mathcal{M}_\theta \cdot \frac{1}{p^2} \int\int \|\boldsymbol{z}-\boldsymbol{w}\|_2^2 |K_q^{(p)}(\boldsymbol{z},\boldsymbol{w})|^2 \mathrm{d}q(\boldsymbol{z})\mathrm{d}q(\boldsymbol{w}), \tag{S4}
$$

where we used the 1-Lipschitzianity of $\frac{\widehat{\nabla_\theta \mathcal{L}}(\boldsymbol{z},\theta)}{q(\boldsymbol{z})K_q^{(p)}(\boldsymbol{z},\boldsymbol{z})}$, with $\mathcal{M}_\theta = \mathcal{O}_P(1)$ the Lipschitz constant.

We control the integral in (S4) by invoking the renowned Christoffel-Darboux formula for the OPE kernel $K_q^{(p)}$ [4]. As outlined in the main text, a fundamental idea which enables us to obtain reduced fluctuations in our determinantal samplers is that, in the context of (S4), the $\|\boldsymbol{z}-\boldsymbol{w}\|_2$ term crucially suppresses fluctuations near the diagonal $\boldsymbol{z}=\boldsymbol{w}$; whereas far from the diagonal, the fluctuations are suppressed by the decay of the OPE kernel $K_q^{(p)}$.

In Sections S3.1.1 and S3.1.2 below, we provide the details of how this idea is implemented by exploiting the Christoffel-Darboux formula; first detailing the simpler 1D case, and subsequently examining the case of general $d$ dimensions. In (S6), we will finally demonstrate the desired $\mathcal{O}_P(p^{-(1+1/d)})$ order of the fluctuations of $\Xi_{A,\mathrm{s}}$ given the dataset $\mathfrak{D}$, thereby completing the proof.

### S3.1.1 Reduced fluctuations in one dimension

We first demonstrate how to control (S4) for $d=1$.

The Christoffel-Darboux formula reads

$$
K_q^{(p)}(x,y) = a_p \cdot \frac{\phi_p(x)\phi_{p-1}(y) - \phi_p(y)\phi_{p-1}(x)}{x-y}, \tag{S5}
$$

where $a_p$ is the so-called first recurrence coefficient of $q$; see [4, Section 1 to 3]. But we assumed in Section 3.3 that $q$ is Nevai-class, which actually implies $a_p \to 1/2$ by definition; see [5, Section 4]. This implies that, in $d=1$, we have

$$
\mathrm{Var}[\Xi_{A,\mathrm{s}}|\mathfrak{D}] \lesssim \mathcal{M}_\theta \cdot \frac{1}{p^2} \cdot \iint (x-y)^2 \cdot \frac{(\phi_p(x)\phi_{p-1}(y) - \phi_p(y)\phi_{p-1}(x))^2}{(x-y)^2} \mathrm{d}q(x)\mathrm{d}q(y)
$$

$$
\leq \mathcal{M}_\theta \cdot \frac{1}{p^2} \cdot \iint (\phi_p(x)\phi_{p-1}(y) - \phi_p(y)\phi_{p-1}(x))^2 \, \mathrm{d}q(x)\mathrm{d}q(y)
$$

$$
= \mathcal{M}_\theta \cdot \frac{1}{p^2} \cdot \left( 2\|\phi_p\|_{L_2(q)}^2 \|\phi_{p-1}\|_{L_2(q)}^2 - 2\langle \phi_p, \phi_{p-1}\rangle^2 \right)
$$

$$
= 2\mathcal{M}_\theta \cdot \frac{1}{p^2}.
$$

### S3.1.2 Reduced fluctuations in general dimension

For $d>1$, following the main text we consider that the measure $q$ splits as a product measure of $d$ Nevai-class $q_i$s, i.e. $q = \otimes_{i=1}^d q_i$. Assume that $p = m^d$ for some $m \in \mathbb{N}$ for simplicity; we discuss at the end of the section how to treat the case $p^{1/d} \notin \mathbb{N}$.

Because of the graded lexicographic ordering that we chose for multivariate monomials, $K_q^{(p)}$ writes as a tensor product of $d$ coordinate-wise OPE kernels of degree $m$, namely $K_q^{(p)}(\cdot,\cdot) = \prod_{j=1}^d K_{q_j}^{(m)}(\cdot,\cdot)$. Now, observe that for any $j = 1, \ldots, d$,

$$
\iint |K_{q_j}^{(m)}(z,y)|^2 \mathrm{d}\gamma(y)\mathrm{d}\gamma(x) = \int K_{q_j}^{(m)}(x,x)\mathrm{d}\gamma(x) = m.
$$

As a result, for $d > 1$, setting $\boldsymbol{z} = (x_1, x_2, \ldots, x_d)$ and $\boldsymbol{w} = (y_1, y_2, \ldots, y_d)$, it comes

$$\mathrm{Var}[\Xi_{A,\mathrm{DPP}}|\mathfrak{D}] \lesssim \mathcal{M}_\theta \cdot \frac{1}{p^2} \cdot \iint \left( \sum_{i=1}^d (x_i - y_i)^2 \right) \cdot |K_q^{(p)}(\boldsymbol{z}, \boldsymbol{w})|^2 \mathrm{d}q(\boldsymbol{z}) \mathrm{d}q(\boldsymbol{w})$$

$$= \mathcal{M}_\theta \cdot \frac{1}{p^2} \cdot \sum_{i=1}^d \left( \iint (x_i - y_i)^2 \cdot |K_q^{(p)}(\boldsymbol{z}, \boldsymbol{w})|^2 \mathrm{d}q(\boldsymbol{z}) \mathrm{d}q(\boldsymbol{w}) \right)$$

$$= \mathcal{M}_\theta \cdot \frac{1}{p^2} \cdot \sum_{i=1}^d \left( \iint (x_i - y_i)^2 \cdot \prod_{j=1}^d |K^j(x_j, y_j)|^2 \prod_{j=1}^d \mathrm{d}q_j(x_j) \mathrm{d}q_j(y_j) \right),$$

leading to

$$\mathrm{Var}[\Xi_{A,\mathrm{DPP}}|\mathfrak{D}]$$

$$= \mathcal{M}_\theta \frac{1}{p^2} \sum_{i=1}^d \left[ \prod_{j \neq i} \left( \iint |K^j(x_j, y_j)|^2 \mathrm{d}q_j(x_j) \mathrm{d}q_j(y_j) \right) \cdot \left( \iint (x_i - y_i)^2 |K^i(x_i, y_i)|^2 \mathrm{d}q_i(x_i) \mathrm{d}q_i(y_i) \right) \right]$$

$$= \mathcal{M}_\theta \cdot \frac{1}{p^2} \cdot \sum_{i=1}^d \left[ (p^{1/d})^{d-1} \cdot \left( \iint (x_i - y_i)^2 |K^i(x_i, y_i)|^2 \mathrm{d}q_i(x_i) \mathrm{d}q_i(y_i) \right) \right]$$

$$\lesssim \mathcal{M}_\theta \cdot \frac{1}{p^2} \cdot \sum_{i=1}^d \left[ (p^{1/d})^{d-1} \cdot 2 \right]$$

$$\lesssim \mathcal{M}_\theta \cdot \frac{d}{p^{1+1/d}}, \tag{S6}$$

where we have used our earlier analyses of Section S3.1.1.

When $p^{1/d} \notin \mathbb{N}$, the kernel $K_q^{(p)}$ does not necessarily decompose as a product of $d$ one-dimensional OPE kernels. However, the graded lexicographic order that we borrowed from [5] ensures that $K_q^{(p)}$ departs from such a product only up to $\mathcal{O}(\lfloor p^{1/d} \rfloor)$ additional terms, whose influence on the variance can be controlled by a counting argument; see e.g. [5, Lemma 5.3].

## S3.2 Fluctuation analysis for $\Xi_{A,\mathrm{DPP}}$

Again using the formula (7) for the variance of linear statistics of DPPs, and remembering that $\widetilde{\mathbf{K}}$ is a projection matrix, we may write

$$\mathrm{Var}[\Xi_{A,\mathrm{DPP}}|\mathfrak{D}] = \frac{1}{N^2} \cdot \sum_{i=1}^N \sum_{j=1}^N \left\| \nabla_\theta \mathcal{L}(\boldsymbol{z}_i, \theta) \widetilde{\mathbf{K}}_{ii}^{-1} - \nabla_\theta \mathcal{L}(\boldsymbol{z}_j, \theta) \widetilde{\mathbf{K}}_{jj}^{-1} \right\|_2^2 |\widetilde{\mathbf{K}}_{ij}|^2. \tag{S7}$$

At this point, we set $\tau$ to be the exponent in the order of spectral approximation $\mathcal{O}_P(N^{-\tau})$ obtained in Section S3.2.1 (our analysis in Section S3.2.1 indicates a choice of $\tau = 1/2$; nonetheless we present the analysis here in terms of the parameter $\tau$, so as to leave open the possibility of simple updates to our bound based on more improved analysis of the spectral approximation). We use the integral and spectral approximations (S15) and (S11), and the inequality $|a - b|^2 \leq 2|a|^2 + 2|b|^2$ to continue from (S7) as

$$\left\| \nabla_\theta \mathcal{L}(\boldsymbol{z}_i, \theta) \widetilde{\mathbf{K}}(\boldsymbol{z}_i, \boldsymbol{z}_i)^{-1} - \nabla_\theta \mathcal{L}(\boldsymbol{z}_j, \theta) \widetilde{\mathbf{K}}(\boldsymbol{z}_j, \boldsymbol{z}_j)^{-1} \right\|_2^2 |\widetilde{\mathbf{K}}(\boldsymbol{z}_i, \boldsymbol{z}_j)|^2$$

$$\leq \left( \left( 2 \left\| \nabla_\theta \mathcal{L}(\boldsymbol{z}_i, \theta) K_{q,\tilde{\gamma}}^{(p)}(\boldsymbol{z}_i, \boldsymbol{z}_i)^{-1} - \nabla_\theta \mathcal{L}(\boldsymbol{z}_j, \theta) K_{q,\tilde{\gamma}}^{(p)}(\boldsymbol{z}_j, \boldsymbol{z}_j)^{-1} \right\|_2^2 \right. \right.$$

$$\left. \left. + (K_{q,\tilde{\gamma}}^{(p)}(\boldsymbol{z}_i, \boldsymbol{z}_i)^{-2} + K_{q,\tilde{\gamma}}^{(p)}(\boldsymbol{z}_j, \boldsymbol{z}_j)^{-2}) \mathcal{O}_P(N^{-2\tau}) \right) \times \left( 2|K_{q,\tilde{\gamma}}^{(p)}(\boldsymbol{z}_i, \boldsymbol{z}_j)|^2 + \mathcal{O}_P(N^{-2\tau}) \right). \tag{S8}$$

We may combine (S7) and (S8) to obtain

$$
\mathrm{Var}[\Xi_{A,\mathrm{DPP}}|\mathfrak{D}]
$$

$$
\lesssim \frac{1}{N^2} \cdot \sum_{i=1}^{N} \sum_{j=1}^{N} \left\| \nabla_\theta \mathcal{L}(z_i,\theta) K_{q,\tilde{\gamma}}^{(p)}(z_i,z_i)^{-1} - \nabla_\theta \mathcal{L}(z_j,\theta) K_{q,\tilde{\gamma}}^{(p)}(z_j,z_j)^{-1} \right\|_2^2 |K_{q,\tilde{\gamma}}^{(p)}(z_i,z_j)|^2
$$

$$
+ \frac{1}{N^2} \cdot \sum_{i=1}^{N} \sum_{j=1}^{N} \mathcal{O}_P(N^{-2\tau}) \cdot \left( \frac{1}{K_{q,\tilde{\gamma}}^{(p)}(z_i,z_i)^2} + \frac{1}{K_{q,\tilde{\gamma}}^{(p)}(z_j,z_j)^2} \right) \cdot |K_{q,\tilde{\gamma}}^{(p)}(z_i,z_j)|^2
$$

$$
+ \frac{1}{N^2} \cdot \sum_{i=1}^{N} \sum_{j=1}^{N} \left\| \nabla_\theta \mathcal{L}(z_i,\theta) K_{q,\tilde{\gamma}}^{(p)}(z_i,z_i)^{-1} - \nabla_\theta \mathcal{L}(z_j,\theta) K_{q,\tilde{\gamma}}^{(p)}(z_j,z_j)^{-1} \right\|_2^2 \mathcal{O}_P(N^{-2\tau})
$$

$$
+ \mathcal{O}_P(N^{-2\tau}). \tag{S9}
$$

This is where we need more assumptions. We recall our assumption that $\nabla_\theta \mathcal{L}(z,\theta)(\frac{1}{p} K_q^{(p)}(z,z))^{-1}$ is uniformly bounded in $z \in D$. This assumption is justified, for the kernel part, by OPE asymptotics for Nevai-class measures; see Totik's classical result [5, Theorem 4.8]. For the gradient part, it is enough to assume that $\nabla_\theta \mathcal{L}(z,\theta)$ is uniformly bounded in $z \in D$ and $\theta$ (with $\gamma(z)$ being bounded away from 0 and $\infty$). Coupled with the hypothesis that $q(z)$ and the density $\gamma(z)$ of $\gamma$ are uniformly bounded away from 0 and $\infty$ on $D$, and the uniform CLT for the convergence $\tilde{\gamma}(z) = \gamma(z) + \mathcal{O}_P(N^{-1/2})$, we may deduce that

$$
\nabla_\theta \mathcal{L}(z,\theta) \left( \frac{1}{p} K_{q,\tilde{\gamma}}^{(p)}(z,z) \right)^{-1}
$$

$$
= \nabla_\theta \mathcal{L}(z,\theta) \left( \frac{1}{p} K_q^{(p)}(z,z) \right)^{-1} \cdot \frac{\tilde{\gamma}(z)}{q(z)}
$$

$$
= \nabla_\theta \mathcal{L}(z,\theta) \left( \frac{1}{p} K_q^{(p)}(z,z) \right)^{-1} \cdot \frac{\gamma(z)}{q(z)} + \nabla_\theta \mathcal{L}(z,\theta)(\frac{1}{p} K_q^{(p)}(z,z))^{-1} \cdot \frac{1}{q(z)} \cdot \mathcal{O}_P(N^{-1/2})
$$

$$
= \mathcal{O}_P(1) + \mathcal{O}_P(N^{-1/2})
$$

$$
= \mathcal{O}_P(1). \tag{S10}
$$

We now demonstrate how the spectral approximations lead to approximations for integrals appearing in the above fluctuation analysis for $\Xi_{A,\mathrm{DPP}}$. To give the general structure of the argument, to be invoked multiple times in the following, we note that

$$
\frac{1}{N^2} \sum_{i,j=1}^{N} \mathcal{O}_P(a(N)) \left| \frac{1}{p} \cdot K_{q,\tilde{\gamma}}^{(p)}(z_i,z_j) \right|^2 = \mathcal{O}_P(a(N)), \tag{S11}
$$

using the fact that $\frac{1}{p} \cdot |K_{q,\tilde{\gamma}}^{(p)}(z_i,z_j)| \le \sqrt{\frac{1}{p} \cdot K_{q,\tilde{\gamma}}^{(p)}(z_i,z_i)} \sqrt{\frac{1}{p} \cdot K_{q,\tilde{\gamma}}^{(p)}(z_j,z_j)}$ via the Cauchy-Schwarz inequality, and that $\frac{1}{p} \cdot K_{q,\tilde{\gamma}}^{(p)}(z_i,z_i)$ is $\mathcal{O}_P(1)$ by OPE asymptotics.

We may combine (S9) and (S10) to deduce that

$$
\mathrm{Var}[\Xi_{A,\mathrm{DPP}}|\mathfrak{D}]
$$

$$
\lesssim \frac{1}{N^2} \cdot \sum_{i=1}^{N} \sum_{j=1}^{N} \left\| \nabla_\theta \mathcal{L}(z_i,\theta) K_{q,\tilde{\gamma}}^{(p)}(z_i,z_i)^{-1} - \nabla_\theta \mathcal{L}(z_j,\theta) K_{q,\tilde{\gamma}}^{(p)}(z_j,z_j)^{-1} \right\|_2^2 |K_{q,\tilde{\gamma}}^{(p)}(z_i,z_j)|^2
$$

$$
+ \mathcal{O}_P(N^{-2\tau}). \tag{S12}
$$

We now proceed as from (S12) as follows:

$$\mathrm{Var}[\Xi_{A,\mathrm{DPP}}|\mathfrak{D}]$$

$$\lesssim \frac{1}{N^2}\sum_{i,j=1}^{N}\left\|\nabla_\theta\mathcal{L}(z_i,\theta)\left(\frac{1}{p}K_{q,\hat\gamma}^{(p)}(z_i,z_i)\right)^{-1}-\nabla_\theta\mathcal{L}(z_j,\theta)\left(\frac{1}{p}K_{q,\hat\gamma}^{(p)}(z_j,z_j)\right)^{-1}\right\|_2^2\left|\frac{1}{p}K_{q,\hat\gamma}^{(p)}(z_i,z_j)\right|^2$$
$$\quad + \mathcal{O}_P(N^{-2\tau})$$

$$\lesssim \iint\left\|\nabla_\theta\mathcal{L}(z,\theta)\left(\frac{1}{p}K_{q,\hat\gamma}^{(p)}(z,z)\right)^{-1}-\nabla_\theta\mathcal{L}(w,\theta)\left(\frac{1}{p}K_{q,\hat\gamma}^{(p)}(w,w)\right)^{-1}\right\|_2^2\left|\frac{1}{p}K_{q,\hat\gamma}^{(p)}(z,w)\right|^2\mathrm{d}\gamma(z)\mathrm{d}\gamma(w)$$
$$\quad + \mathcal{O}_P(N^{-1/2}) + \mathcal{O}_P(N^{-2\tau})$$

$$\lesssim \iint\left\|\nabla_\theta\mathcal{L}(z,\theta)\left(\frac{1}{p}K_{q,\tilde\gamma}^{(p)}(z,z)\right)^{-1}-\nabla_\theta\mathcal{L}(w,\theta)\left(\frac{1}{p}K_{q,\tilde\gamma}^{(p)}(w,w)\right)^{-1}\right\|_2^2\left|\frac{1}{p}K_{q,\tilde\gamma}^{(p)}(z,w)\right|^2\mathrm{d}\tilde\gamma(z)\mathrm{d}\tilde\gamma(w)$$
$$\quad + \mathcal{O}_P(N^{-1/2}) + \mathcal{O}_P(N^{-2\tau})$$

$$\sim \frac{1}{p^2}\iint\left\|\nabla_\theta\mathcal{L}(z,\theta)\left(\frac{1}{p}K_{q,\tilde\gamma}^{(p)}(z,z)\right)^{-1}-\nabla_\theta\mathcal{L}(w,\theta)\left(\frac{1}{p}K_{q,\tilde\gamma}^{(p)}(w,w)\right)^{-1}\right\|_2^2\left|K_q^{(p)}(z,w)\right|^2\mathrm{d}q(z)\mathrm{d}q(w)$$
$$\quad + \mathcal{O}_P(N^{-1/2}) + \mathcal{O}_P(N^{-2\tau}), \tag{S13}$$

where, in the last two steps, we have used the assumption that $\left|\frac{1}{p}K_{q,\tilde\gamma}^{(p)}(z,w)\right|^2$ and $\nabla_\theta\mathcal{L}(z,\theta)\left(\frac{1}{p}K_{q,\tilde\gamma}^{(p)}(z,z)\right)^{-1}$ are $\mathcal{O}_P(1)$, and used the rate of convergence of $\hat\gamma_N$ to $\gamma$ to pass from the sum to the integral respect to $\gamma$, and from there to the integral with respect to $\tilde\gamma$ using the rate estimates for kernel density estimation, incurring an additive cost of $\mathcal{O}_P(N^{-1/2})$ in each step.

Using the hypothesis that $\nabla_\theta\mathcal{L}(z,\theta)\left(\frac{1}{p}K_{q,\tilde\gamma}^{(p)}(z,z)\right)^{-1}$ is 1-Lipschitz with a Lipschitz constant that is $\mathcal{O}_P(1)$, we may proceed from (S13) as

$$\mathrm{Var}[\Xi_{A,\mathrm{DPP}}|\mathfrak{D}]\lesssim \mathcal{M}_\theta\frac{1}{p^2}\int\int\|z-w\|_2^2\left|K_q^{(p)}(z,w)\right|^2\mathrm{d}q(z)\mathrm{d}q(w)+\mathcal{O}_P(N^{-1/2})+\mathcal{O}_P(N^{-2\tau}),$$
$$\tag{S14}$$

where $\mathcal{M}_\theta^{1/2}$ is the Lipschitz constant that is $\mathcal{O}_P(1)$. We are back to analyzing the same variance term as in Section S3.1, and the rest of the proof follows the very same lines.

### S3.2.1 Spectral approximations

We analyse in this section the approximation error when we replace $K_{q,\tilde\gamma}^{(p)}$ by $\widetilde{\mathbf{K}}$ in Equation (S8). To this end, we study the difference between the entries of $\widetilde{\mathbf{K}}$ and those of $K_{q,\tilde\gamma}^{(p)}(\cdot,\cdot)$ when restricted to the data set $\mathfrak{D}$. We recall the fact that $\widetilde{\mathbf{K}}$ is viewed as a kernel on the space $L_2(\hat\gamma_N)$.

The idea is that, since $N$ is large, the kernel $\widetilde{\mathbf{K}}$ acting on $L_2(\hat\gamma_N)$, which is obtained by spectrally rounding off the kernel $K_{q,\tilde\gamma}^{(p)}$ acting on $L_2(\hat\gamma_N)$, is well approximated by the kernel $K_{q,\tilde\gamma}^{(p)}$ acting on $L_2(\tilde\gamma)$. By definition, $K_{q,\tilde\gamma}^{(p)}(z,w) = \sqrt{\frac{q(z)}{\tilde\gamma(z)}}K_q^{(p)}(z,w)\sqrt{\frac{q(w)}{\tilde\gamma(w)}}$, we may deduce that $K_{q,\tilde\gamma}^{(p)}(\cdot,\cdot)\mathrm{d}\tilde\gamma(\cdot)\mathrm{d}\tilde\gamma(\cdot) = K_q^{(p)}(\cdot,\cdot)\mathrm{d}q(\cdot)\mathrm{d}q(\cdot)$. Now, the kernel $K_q^{(p)}(\cdot,\cdot)$ is a projection on $L_2(q)$. As such, the spectrum of $(K_{q,\tilde\gamma}^{(p)},\mathrm{d}\tilde\gamma)$ is also close to a projection. Since $\widetilde{\mathbf{K}}$ is obtained by rounding off the spectrum of $K_{q,\tilde\gamma}^{(p)}$ to $\{0,1\}$, the quantities $|\widetilde{\mathbf{K}}(z_i,z_j) - K_{q,\tilde\gamma}^{(p)}(z_i,z_j)|$ will be ultimately by controlled by how close the kernel $(K_{q,\tilde\gamma}^{(p)}|_{\mathfrak{D}},\hat\gamma_N)$ is from a true projection.

To analyse this, we consider the operator $[K_{q,\tilde\gamma}^{(p)}|_{\mathfrak{D}}]^2$ on $L_2(\hat\gamma_N)$, which is an integral operator given by the convolution kernel

$$K_{q,\tilde\gamma}^{(p)}|_{\mathfrak{D}} \star K_{q,\tilde\gamma}^{(p)}|_{\mathfrak{D}}(\boldsymbol{z}_i, \boldsymbol{z}_k) = \int K_{q,\tilde\gamma}^{(p)}(\boldsymbol{z}_i, \boldsymbol{z}_j) K_{q,\tilde\gamma}^{(p)}(\boldsymbol{z}_j, \boldsymbol{z}_k) \mathrm{d}\hat\gamma_N(\boldsymbol{z}_j)$$

$$= \frac{1}{N} \sum_{j=1}^{N} K_{q,\tilde\gamma}^{(p)}(\boldsymbol{z}_i, \boldsymbol{z}_j) K_{q,\tilde\gamma}^{(p)}(\boldsymbol{z}_j, \boldsymbol{z}_k)$$

$$= \int K_{q,\tilde\gamma}^{(p)}(\boldsymbol{z}_i, \boldsymbol{w}) K_{q,\tilde\gamma}^{(p)}(\boldsymbol{w}, \boldsymbol{z}_k) \mathrm{d}\tilde\gamma(\boldsymbol{w}) + \mathcal{O}_P(N^{-1/2})$$

$$= K_{q,\tilde\gamma}^{(p)}(\boldsymbol{z}_i, \boldsymbol{z}_k) + \mathcal{O}_P(N^{-1/2})$$

$$= K_{q,\tilde\gamma}^{(p)}|_{\mathfrak{D}}(\boldsymbol{z}_i, \boldsymbol{z}_k) + \mathcal{O}_P(N^{-1/2}),$$

where we have used the convergence of the kernel density estimator $\tilde\gamma$ as well as the empirical measure $\hat\gamma_N$ to the underlying measure $\gamma$, at the rate $\mathcal{O}_P(N^{-1/2})$ described e.g. by the uniform CLT.

We may summarize the above by observing that $K_{q,\tilde\gamma}^{(p)}|_{\mathfrak{D}}^2$ on $L_2(\hat\gamma_N)$ is a projection up to an error of $\mathcal{O}_P(N^{-1/2})$, which indicates an approximation of $|\widetilde{\mathbf{K}}(\boldsymbol{z}_i, \boldsymbol{z}_j) - K_{q,\tilde\gamma}^{(p)}|_{\mathfrak{D}}(\boldsymbol{z}_i, \boldsymbol{z}_j)| = \mathcal{O}_P(N^{-1/2})$.

To understand the estimator $\Xi_{A,\mathrm{DPP}}$, we also need to understand $\widetilde{\mathbf{K}}(\boldsymbol{z}_i, \boldsymbol{z}_i)^{-1}$, which we will deduce from the above discussion. To this end, we observe that

$$|\widetilde{\mathbf{K}}(\boldsymbol{z}_i, \boldsymbol{z}_i)^{-1} - K_{q,\tilde\gamma}^{(p)}(\boldsymbol{z}_i, \boldsymbol{z}_i)^{-1}| = |(K_{q,\tilde\gamma}^{(p)}(\boldsymbol{z}_i, \boldsymbol{z}_i) + \mathcal{O}_P(N^{-1/2}))^{-1} - K_{q,\tilde\gamma}^{(p)}(\boldsymbol{z}_i, \boldsymbol{z}_i)^{-1}|$$

$$= K_{q,\tilde\gamma}^{(p)}(\boldsymbol{z}_i, \boldsymbol{z}_i)^{-1} \cdot \mathcal{O}_P(N^{-1/2}). \tag{S15}$$

In drawing the above conclusion, we require that $K_{q,\tilde\gamma}^{(p)}(\boldsymbol{z}_i, \boldsymbol{z}_i)$ stays bounded away from 0, which we justity as follows. We recall that $K_{q,\tilde\gamma}^{(p)}(\boldsymbol{z}_i, \boldsymbol{z}_i) = K_q^{(p)}(\boldsymbol{z}_i, \boldsymbol{z}_i) \cdot \frac{q(\boldsymbol{z}_i)}{\tilde\gamma(\boldsymbol{z}_i)}$. We recall from OPE asymptotics that $K_q^{(p)}(\boldsymbol{z}_i, \boldsymbol{z}_i)$ is of the order $p$; whereas $\tilde\gamma(\boldsymbol{z}_i) = \gamma(\boldsymbol{z}_i) + \mathcal{O}_P(N^{-1/2})$ from kernel density approximation and uniform CLT asymptotics. We recall our hypothesis that the densities $q, \gamma$ are bounded away from 0 and $\infty$ on $\mathfrak{D}$. Putting together all of the above, we deduce that $K_{q,\tilde\gamma}^{(p)}(\boldsymbol{z}_i, \boldsymbol{z}_i)$ is of order $p$; in particular it is bounded away from 0 as desired.

### S3.3 Order of $p$ vs $N$ and future work

In this section, we discuss the relative order of $p$ and $N$, especially in the context of the estimator $\Xi_{A,\mathrm{s}}$. In order to do this for $\Xi_{A,\mathrm{s}}$, we undertake a classic bias-variance trade-off argument. To this end, we recall that while the bias is $\mathcal{O}_P(ph/N)$ (with $h$ being the window size for kernel smoothing), the variance is $\mathcal{O}_P(p^{-(1+1/d)})$. Further, we substitute one of the canonical choices for the window size $h$, which is to set $h = N^{-1/d}$ for dimension $d$ and data size $N$. Setting the bias and the standard deviation to be roughly of the same order, we obtain a choice of $p$ as $p = N^{\frac{2d+1}{3d+1}}$. For the estimator $\Xi_{A,\mathrm{DPP}}$, a similar analysis may be undertaken. However, while the finite sample bias is 0, the variance term is more complicated, particularly with the contributions from the spectral approximation $\mathcal{O}_P(N^{-\tau})$. We believe that the present analysis of the spectral approximation can be further tightened and rigorised to yield more optimal values of $\tau$ that more closely mimic experimental performance. Further avenues of improvement include better ways to handle the boundary effects (to control the asymptotic intensity of the OPE kernels that behave in a complicated manner at the boundary of the background measure); methods to bypass the spectral round-off step in constructing the estimator $\Xi_{A,\mathrm{DPP}}$; hands-on analysis of the errors caused by switching between discrete measures and continuous densities that is tailored to our setting (and therefore raising the possibility of sharper error bounds), among others.

## S4 Experiments on a real dataset

To extensively compare the performance of our gradient estimator $\Xi_{A,\mathrm{DPP}}$ to the default $\Xi_{A,\mathrm{Poi}}$, we run the same experiment as in Section 5 of the main paper, but on a benchmark dataset from

LIBSVM[3]. We download the *letter* dataset, which consists of 15000 training samples and 5000 testing samples, where each sample contains 16 features. We modify the 26-class classification problem into a binary classification problem where the goal is to separate the classes 1-13 from the other 13 classes. Denote the preprocessed dataset as *letter.binary*. We consider $\mathcal{L} = \mathcal{L}_{\text{lin}}$, $\lambda_0 = 0.001$ and $p = 10$. Figure 1 summarizes the experimental results on *letter.binary*, where the performance metrics are averaged over 1000 independent runs of each SGD variant. The left figure shows the decrease of the objective function value, the middle figure shows how the norm of the complete gradient $\|\Xi_N(\theta_t)\|$ decreases with the *budget*, and the right figure shows the value of the test error. Error bars in the last figure are $\pm$ one standard deviation of the mean. In the experiment, we can see that using a DPP improves over Poisson minibatches of the same size both in terms of minimizing the empirical loss, and of reaching a small test error with a given budget. Compared to the experimental results on the simulated data in the main paper, we can see that although the $\mathcal{O}_P(p^{-(1+1/d)})$ rate discussed in Section 4 becomes slower as $d$ grows, our DPP-based minibatches still gives better performance on this real dataset with $d = 16$ compared to Poisson minibatches of the same size, which again demonstrates the significance of variance reduction in SGD.

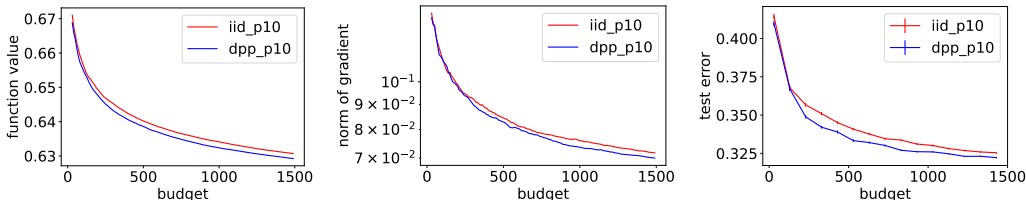

Figure 1: Logistic regression on the *letter.binary* dataset.