# OpenReview forum: "Determinantal point processes based on orthogonal polynomials for sampling minibatches in SGD"
_NeurIPS.cc/2021/Conference — NeurIPS 2021 Spotlight_

### Official Review · Reviewer_7px8 · 2021-07-09

**Rating:** 8
**Confidence:** 3

**Summary:**

This paper presents a framework for variance reduction in stochastic gradient descent (SGD) that involves detrimental point processes (DPPs) based on orthogonal polynomials.  This framework uses a class of continuous DPPs based on orthogonal polynomials, and kernel density estimators built from the data, to perform minibatch sampling in SGD.  Theoretical analysis of this approach proves a bound on the variance for two gradient estimators, leading to variance reduction compared to uniform minibatch sampling without replacement.  This approach is empirically validated using experiments performed on synthetic simulated data.

**Limitations And Societal Impact:**

The authors have included a short, but adequate, discussion of the limitations of their work in Section 6 of the paper.  Since this work involves very general algorithmic and theoretical contributions, a discussion of potential negative societal impact does not appear to be applicable.

**Main Review:**

The minibatch sampling approach presented in this paper appears to be novel and technically sound, and the paper is reasonably well written.  The main theoretical result involving the bound on the gradient variance appears to be significant and will likely serve as a solid basis for future work in this area.  This does indeed seem to be one of the first theoretical results regarding the variance reduction that results from DPP-based minibatch sampling.  The sketch of the proof for Proposition 4 (the main theoretical contribution, regarding the variance bound) in Section 4.2 is reasonably clear.  However, this proof involves a fair amount of technical complexity and background, and I did not fully check the proof.

A few aspects of this paper could be improved.  First, while the presentation of DPPs based on orthogonal polynomials, called multivariate orthogonal polynomial ensembles (OPEs), is reasonably clear, the motivation behind the choice of OPEs in this approach is not very clear.  It would be helpful for the authors to add a discussion regarding why OPEs are useful for variance reduction, and the intuition behind OPEs, to the beginning of the paper.  Second, experimental results on a real dataset are found only in the appendix.  It would be better to include these real-dataset results in the main paper, along with a discussion regarding the relative differences in the extent of variance reduction as compared to the simulated data.


**Time Spent Reviewing:**

5 hours

---

> ### Author Response · Authors · 2021-08-10
> **Response to Reviewer 7px8**
>
> Thank you for your review.
>
> > First, while the presentation of DPPs based on orthogonal polynomials, called multivariate orthogonal polynomial ensembles (OPEs), is reasonably clear, the motivation behind the choice of OPEs in this approach is not very clear. It would be helpful for the authors to add a discussion regarding why OPEs are useful for variance reduction, and the intuition behind OPEs, to the beginning of the paper.
>
> We thank the reviewer for this suggestion, and will add a few words on this topic. Roughly speaking, we would like to use a DPP that is tailored to the data distribution at hand. OPEs provide a natural way of associating a DPP to a given measure, along with a substantive body of mathematical literature and tools that can be summoned as per necessity. This makes it a natural choice for our purposes. However, this does not preclude other possibilities of DPP construction, which can be explored in follow-up work.
>
> > Second, experimental results on a real dataset are found only in the appendix. It would be better to include these real-dataset results in the main paper, along with a discussion regarding the relative differences in the extent of variance reduction as compared to the simulated data.
>
> We can add the suggested discussion. As for moving the experimental results on the real dataset, we may not have enough room within the page limit. If we must choose, we would prefer keeping the toy experiments over the real dataset in the main text. Indeed, they serve as a controlled validation of the theoretical analysis, the latter being our main contribution.

---

> > ### Comment · Reviewer_7px8 · 2021-08-17
> > **Rebuttal response**
> >
> > Thank you for the response to my questions/comments.  I’ve read through the other reviews and rebuttal comments, and am satisfied with the remarks.  In particular, I think the authors have adequately addressed the questions and issues raised by reviewers KigW and mkKg.  Therefore, I maintain my review score of 8 (Top 50% of accepted NeurIPS papers, clear accept).

---

### Official Review · Reviewer_PADH · 2021-07-16

**Rating:** 7
**Confidence:** 4

**Summary:**

The authors introduce the first complete theoretical analysis of minibatch sampling using determinantal point processes in the context of (batched) stochastic gradient descent.

The core of this analysis is built upon continuous DPP results (specifically, on orthogonal polynomial ensembles, or OPEs) in which it has been shown that DPPs achieve faster-than-iid error decay. Assuming knowledge of a KDE $\tilde \gamma$ for the distribution over data points, the authors introduce two batched estimators of the full gradient:
1) A weighted sum of gradients at points sampled by a DPP with kernel $K$, where $K$ is build by restricting the eigenvalues of an OPE kernel reweighted by $\tilde \gamma$. This kernel (and associated DPP) can be computed and sampled from efficiently.
2) An estimator of more theoretical nature, which can sample batches of points that do not arise in the actual data, and is as least as costly in practice as estimating the full gradient.

The authors then prove that the first estimator is unbiased, and obtain a bound on the bias of the second. They then also show that the variance of the resulting estimator decays at a rate $\le p^{-(1 + \frac1d)}$, where $p$ is an integer parameter controlling the number of polynomials used to build the OPE and $d$ is the dimensionality of the input data.

Finally, the authors show on synthetic data that the resulting samplers improve the empirical convergence speed of SGD as a function of the number of required gradient evaluations, and confirm empirically the theoretical variance decay rate obtained earlier.

**Limitations And Societal Impact:**

No negative societal impact.

**Main Review:**

Originality: This paper is original, and combines a novel analysis with previous results on OPE DPPs and empirical analysis of DPP sampling. This paper provides the first complete analysis of the variance decay of gradient approximations using DPP batched sampling, a question which has been the focus of several works in the past (which are cited).

Quality: This paper is of high quality, well positioned within the relevant literature, and introduces results of extreme relevance to the machine learning community.

Clarity: This paper is very clearly written, despite the complexity of the topic. The authors provide proof sketches that are sufficient to build an intuition for the proofs themselves, carefully tie in their work with previous results, and are upfront about the requirements and limitations of their methods. Their empirical analysis is also careful to include a comparison based on the number of gradient evaluations, an important distinction in this type of analysis.

Significance: This paper is of high significance. The question of using DPPs and other negatively dependent measures to improve convergence rates of batched SGD has been raised several times in previous work by the community, and this paper provides the first theoretical analysis of DPP sampling for minibatch SGD.

I would've liked to see a comparison to importance sampling.

Questions & clarifications
- Related work using DPPs and/or importance sampling to sample minibatches has also looked at building distributions over the data that are dependent on the gradients of the model at the current iteration. Do your results preclude this as a promising direction, are they independent, or could both approaches be combined in the future?

- Section 2: you state the assumption that $\Xi_A$ does not depend on the past of the Markov chain. Under this assumption, does this mean that individual gradients aren't affected by the past of the chain (in the sense that, in practice, the points that were sampled in a previous round of SGD change the model, and as such affect future gradients)?

- I would recommend adding a complexity analysis of the use of the first estimator, including sampling costs (additional to the $\mathcal O(Np^2)$ operations required to build the a rank-$p$ projection DPP. I also believe that readers will be interested in some empirical numbers of the runtime of SGD with the DPP batch sampler as a function of $p$.


**Time Spent Reviewing:**

4 hours

---

> ### Author Response · Authors · 2021-08-10
> **Response to Reviewer PADH**
>
> Thank you for your review.
>
> > I would've liked to see a comparison to importance sampling.
>
> This is an interesting suggestion. However, we mention that the choice of the sample points would still be independent (albeit with different probabilities), and as such, we do not expect any change in the exponent on p in the variance (i.e., the variance should still be O(1/p) as in uniform sampling). The leading constant in front of the 1/p can possibly be reduced by tuning the importance weights, but an improvement in the exponent (as we have in the DPP-based sampler) is perhaps more consequential. That being said, in line with your comment, given a set of good importance weights, there could be a way of designing a DPP with the same marginal probabilities and a provably fast variance decay. A related problem is considered in [15] in the more difficult setting of coreset sampling, for which no fast variance decay has been proven yet.
>
> > Related work using DPPs and/or importance sampling to sample minibatches has also looked at building distributions over the data that are dependent on the gradients of the model at the current iteration. Do your results preclude this as a promising direction, are they independent, or could both approaches be combined in the future?
>
> This is an excellent suggestion, and our results do not preclude it at all. Our present work is arguably the first rigorous result in this direction, and we certainly plan to explore such finer extensions of this approach in forthcoming work.
>
> > Section 2: Under this assumption, does this mean that individual gradients aren't affected by the past of the chain (in the sense that, in practice, the points that were sampled in a previous round of SGD change the model, and as such affect future gradients)?
>
> Indeed, they are not affected. The reviewer is right.
>
> > I would recommend adding a complexity analysis of the use of the first estimator, including sampling costs (additional to the
> operations required to build the a rank-p projection DPP. I also believe that readers will be interested in some empirical numbers of the runtime of SGD with the DPP batch sampler as a function of $p$.
>
> We will add the detail of the complexity of the first estimator and runtimes. In our toy experiments, the runtime (around a millisecond per iteration) is clearly dominated by the actual $O(Np^2)$ DPP sampling step. This is an effect of using Python for DPP sampling while all other steps mostly involve inner products and are thus efficiently treated by NumPy. For more sophisticated (e.g. highly non-linear) models and with a comparable implementation of all steps in the same language, the bottleneck will shift from DPP sampling,which remains $O(Np^2)$, to individual gradient evaluations (which become arbitrarily expensive with model complexity). See also the second item in our answer to Reviewer KigW.

---

> > ### Comment · Reviewer_PADH · 2021-08-26
> > **Rebuttal response**
> >
> > Thank you for your reply and your answers to my questions (as well as those of the other reviewers)! I am maintaining my score of 7.

---

### Official Review · Reviewer_mkKg · 2021-07-17

**Rating:** 6
**Confidence:** 4

**Summary:**

This paper studies unbiased estimators of SGD minibatch via DPP with orthogonal polynomials. The estimator can be constructed by sampling DPP whose kernel consists of the orthogonal projection of some kernel proposition. Samples from the projection DPP are guaranteed fixed sizes (corresponds to the rank) and the variance of minibatch SGD can be achieved by O(p^{-(1 + 1/d)}) for batch size p and feature dimension d. This is smaller than the uniform batch sampling with O(p^{-1}) which leads us that DPP can play a role of variance reduction. Empirical results on both synthetic and real-world datasets show the superior performance of DPP sampling than uniform one in terms of the norm of gradients, distance to the optimal solution as well as the variance of gradient estimations.

**Limitations And Societal Impact:**

Although DPPs can be good at reducing the variance of gradient estimators, their computations have always been critical issues and many works have tried to resolve them. More precisely, the proposed method can reduce the variance O(p^{-1/d}) time better than uniform sampling, but the time to sample mini-batches from a DPP is O(N*p^2) in general. Hence, in terms of computational time, DPP may have no advantage over uniform sampling.

These days, machine learning models have become more complex (likely to be deep architecture) and outperform linear models. However, the proposed method seems valid for linear models. Does it have a chance to extend more complex models, e.g., deep architectures?


**Main Review:**

DPPs have been used for minibatch selection in SGD as a variance reduction. This work proposes a careful analysis of unbiased estimator of minibatch SGD based on DPP with an orthogonal polynomial. It is interesting that feature dimension d actually matters the variance of estimators rather than distribution of data points. However, if the data distribution is uniform then it seems that DPP sampling is equivalent to uniform sampling. In this case, it is still unclear that Proposition 4 still holds.

**Time Spent Reviewing:**

4

---

> ### Author Response · Authors · 2021-08-10
> **Response to Reviewer mkKg**
>
> Thank you for your review.
>
> > However, if the data distribution is uniform then it seems that DPP sampling is equivalent to uniform sampling. In this case, it is still unclear that Proposition 4 still holds. (...)
>
> We believe there is an important misunderstanding here. Our DPP sampling is *not* uniform sampling, even if the data distribution is uniform. The data distribution being uniform will merely result in our DPP being related to (a discretization of) the Orthogonal Polynomial Ensemble DPP for the uniform measure on the unit cube. This is a non-trivial DPP with actual negative correlations, and definitely not uniform sampling.
>
> > in terms of computational time, DPP may have no advantage over uniform sampling
>
> For complex models, gradient evaluations become the bottleneck, and the $O(Np^2)$ price of sampling our DPP becomes a small price for a smaller variance. Please see the second item in our answer to Reviewer KigW for more details.
>
> > machine learning models have become more complex (likely to be deep architecture) and outperform linear models. However, the proposed method seems valid for linear models. Does it have a chance to extend more complex models, e.g., deep architectures?
>
> We believe there is a misunderstanding here. The proposed method is applicable to any model or protocol which involves stochastic gradient descent, and is not limited nor related to linear models. Our method provides an estimator with desirable fluctuations to approximate the gradient in each step of SGD, and thus is of generic value in all such settings, including deep learning.

---

> > ### Comment · Reviewer_mkKg · 2021-08-18
> > **Follow up**
> >
> > Thanks to authors for clarifying my questions and comments . I missed the point that the repulsive property of DPP makes different from the uniform sampling even the data distribution is uniform. Also, I definitely agree that the proposed method is applicable to any model with SGD, but the point is that the paper would be much welcomed if experiments provide results under more complex models.
> >
> > I have read authors answer to Reviewer KigW regarding the time complexity. In the response, they answered that the gradient evaluations become dominate the DPP sampling of $O(Np^2)$ operations. However, unlike the DPP sampling, the gradient evaluations are not effected by the number of the entire data $N$. It seems that the complexity of DPP sampling cannot be negligible when $N$ is comparable to the number of model parameters. This is a fairly common setting in practice. I think this can be still a limitation of this work unless fast sampling algorithm is not adopted (please correct me I missed). However, I agree that the results of DPP with mini batch SGD are important and all analyses are complete and technically clear. I will change my score to 6.

---

### Official Review · Reviewer_KigW · 2021-07-18

**Rating:** 6
**Confidence:** 2

**Summary:**

The authors introduce a sampling minibatches method in SGD based on determinantal point processes (DPPs) to reduce the estimation variance. Specifically, the authors proposed two gradient estimators using multivariate orthogonal polynomial ensembles, and provide theoretical results on the bias and the variance of the estimators. One of the two proposed estimators is tested on a synthetic dataset.



**Limitations And Societal Impact:**

The paper does not contain this part.

**Main Review:**

The studied problem of reducing SGD estimator variance is very interesting and important. As far as I understand, the proposed method based on DPPs is reasonable and the associated theoretical guarantees are novel. I have the following questions.

1.	It will be better to present the variance result of the first estimator in the main text. Since this estimator is the one that can be used in practice (the experiment section only has results from this estimator). It is not clear to me whether this estimator has the same variance as the second one.
2.	The experiment considers the budget in terms of the number of gradient evaluations. However the cost of each gradient evaluation is different for different methods. Could the authors compare in terms of the runtime? I wonder how much cost the proposed gradient estimator will add to each evaluation.
3.	As the authors mentioned in the relevant literature, there are existing DPPs-based gradient estimators. How will the proposed estimator compare to other DPPs-based estimators empirically? The current experiment only includes Poisson minibatches as the baseline.

In conclusion, I think the proposed method is technically sound and could have a potential impact, but it needs some clarification on the theoretical result and improvement on the empirical results.


**Time Spent Reviewing:**

5

---

> ### Author Response · Authors · 2021-08-10
> **Response to Reviewer KigW**
>
> Thank you for your review.
>
> > It will be better to present the variance result of the first estimator in the main text (...) It is not clear to me whether this estimator has the same variance as the second one.
>
> The first estimator has the same variance as the second one, as discussed L248--252 and in Appendix S3 (Note that there is a typo L251, and that the estimator in the conditional variance should read $\Xi_{A,\text{DPP}}$). But the proof involves additional technical complications and a lengthier argument for variance reduction. When deciding on the paper's layout and because our main contribution was to find a way to prove the variance reduction, we finally decided to put the second estimator in front, to facilitate the exposition and potential reuse of the core theoretical ideas as presented in the current main text.
>
> > The experiment considers the budget in terms of the number of gradient evaluations. However the cost of each gradient evaluation is different for different methods. Could the authors compare in terms of the runtime?
>
> Good point. In our (small-scale) toy experiments, the cost of one iteration is dominated (by two orders of magnitude) by the O(Np^2) sampling of the underlying DPP, which takes around $10^{-3}$ second for $N=1000$, $d=2$ and $p=10$ in our Python implementation. Even the preprocessing steps cost less, e.g. building the kernel density estimator or precomputing the weights in (11). Since our toy models are simple (linear regression and logistic regression), evaluating gradients is very fast using a library like NumPy. A runtime comparison would thus be favourable to i.i.d. sampling, or actually even to vanilla (i.e. non-stochastic) gradient descent, but this would not necessarily be fair or representative of how DPPs can help.
>
> Indeed, as the scale of the data and/or the complexity of the models increase, the bottleneck will shift from DPP sampling (which remains $O(Np^2))$ to individual gradient evaluations. Our comparisons in terms of budget are meant to represent that more realistic regime. For instance, not even mentioning deep nets, learning the parameters of a structured model like a conditional random field leads to arbitrarily costly individual gradients, as the underlying graph gets more dense. Since our focus in the experimental section was to validate the fast rates obtained in the theoretical part and explore the effect of hyperparameters like the batchsize, we limited ourselves to simple experiments. Showing comparisons in terms of budget then allows the reader to extrapolate to more sophisticated models.
>
> That being said, an additional point in favour of DPP-minibatches is that evaluating our gradient estimators does not necessarily require to sample from any DPP. In future work, we will investigate using inverse Laplace sampling of linear statistics of DPPs [24], which can potentially further scale down the $O(Np^2)$ cost using Nyström approximations; see lines 152--175.
>
> > As the authors mentioned in the relevant literature, there are existing DPPs-based gradient estimators. How will the proposed estimator compare to other DPPs-based estimators empirically?
>
> Using a DPP out of the box (say, an RBF kernel) is not guaranteed to yield variance reduction, and one can even cook up datasets and/or kernels that make the resulting variance larger than under Poissonian sampling. This can be seen, e.g., from the proof of Theorem 1 in [4].

---

### Decision · Program_Chairs · 2021-09-27

**Decision:**

Accept (Spotlight)

**Comment:**

This paper presents a convincing theoretical analysis of a technique for minibatch sampling using determinantal point processes, showing that it can lead to variance decaying more rapidly compared with uniform sampling. The result formalizes and refines a commonly felt intuition, and is compelling, significant, and a great fit for NeurIPS. After author feedback, all reviewers favored acceptance.